# Is your video language model a reliable judge?

**Ming Liu, Wensheng Zhang**
Department of Computer Science, Iowa State University
{pkulium,wzhang}@iastate.edu

## Abstract

As video language models (VLMs) gain more applications in various scenarios, the need for robust and scalable evaluation of their performance becomes increasingly critical. The traditional human expert-based evaluation of VLMs has limitations in consistency and scalability, which sparked interest in automatic methods such as employing VLMs to evaluate VLMs. However, the reliability of VLMs as judges remains underexplored. Existing methods often rely on a single VLM as the evaluator. However, this approach can be unreliable or biased because such a model may lack the ability to fully understand the content and may have inherent biases, ultimately compromising evaluation reliability. A remedy is to apply the principle of collective thoughts, aggregating evaluations from multiple VLMs to enhance reliability. This study investigates the efficacy of such approaches, particularly when the pool of judges includes both reliable and unreliable models. Our findings reveal that incorporating collective judgments from such a mixed pool does not necessarily improve the accuracy of the final evaluation. The inclusion of less reliable judges can introduce noise, undermining the overall reliability of the outcomes. To explore the factors that impact evaluation reliability, we fine-tune an underperforming VLM judge, Video-LLaVA, and observe that improved understanding ability alone is insufficient to make VLM judges more reliable. These findings stress the limitations of collective thought approaches and highlight the need for more advanced methods that can account for the reliability of individual models. Our study promotes the development of more reliable evaluation methods for VLMs.

## 1 Introduction

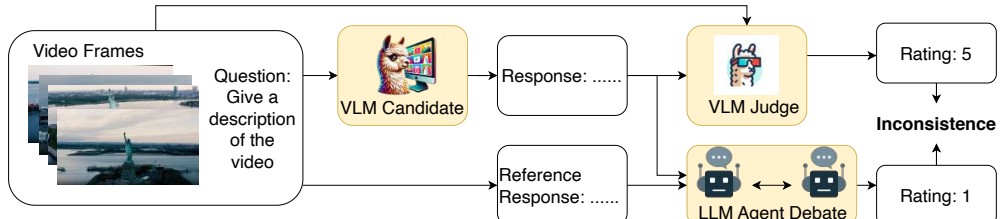

Figure 1: By contrasting reviewing results from one VLM with multi-LLM Agent-Debate, we find that some VLMs are far from being able to provide reliable reviews.

More and more video language models (VLMs) have been developed for video understanding. Given video inputs, these models can generate descriptions and analyses. They have also been adapted to various other tasks, such as evaluating text-to-video generation (Wu et al., 2024; Sun et al., 2025) and providing reward signals in Reinforcement Learning from AI Feedback (Lee et al., 2024; Ahn et al., 2024). Along with this trend, the need for robust and scalable evaluation of VLMs' performance is becoming increasingly critical. Traditionally, the evaluation of VLMs has been performed by human experts whose subjective judgments, however, may be inconsistent and not easily scalable. This limitation has recently sparked interest in automating the evaluation through advanced

machine learning models, specifically employing VLMs to judge other VLMs. However, the reliability of VLMs as judges is underexplored. As illustrated in Figure 1, the evaluation results from a VLM can be significantly inconsistent with those produced by the more reliable reference-guided multi-agent debate method. This is because individual models can exhibit biased outcomes influenced by their training data and architectural constraints, and can be susceptible to hallucinations — generating plausible but incorrect information (Tong et al., 2024; Gervi et al., 2024).

An intuitive solution is to employ the ensemble methods or the principles from *collective intelligence*, aggregating judgments from multiple models to improve reliability, as the concept of collective intelligence suggests that integrating diverse thoughts could improve decision accuracy and mitigate individual biases (Malone et al., 2009).

In order to study in depth the above limitation in evaluating VLMs and the approaches of collective intelligence, we put forward the following research questions:

- Are current VLMs reliable for evaluating VLMs, especially in understanding complex videos? Can we use weaker VLMs to evaluate stronger VLMs?
- Does incorporating collective thought from multiple VLMs enhance the reliability of evaluations?
- What are the limitations of collective thought approaches in the context of VLM evaluation and how can we address them?

To answer these questions, we explore the efficacy of the collective thought approaches in evaluating VLMs. Specifically, we investigate whether pooling judgments across multiple VLMs improves evaluation reliability when the pool of judges includes both reliable and unreliable models, and demonstrate that such a mixed pool does not necessarily improve the reliability. The less reliable models may inject noise into the results that can easily swamp any aggregation benefits. These observations highlight the limitations of collective thought approaches when indiscriminately aggregating evaluations from models of varying reliability. This further underscores the need for more sophisticated methods that can effectively account for the reliability of individual judges and mitigate the influence of less reliable models. The main contributions of our work are summarized as follows:

- We assess the reliability of current VLMs in evaluating VLMs, highlighting their limitations due to inability to fully understand the content or inherent biases.
- We find that using weaker VLMs to judge stronger models leads to unreliable evaluations, as weaker models lack the necessary understanding and critical reasoning abilities.
- We demonstrate that the collective thought approaches, which aggregate judgments from multiple VLMs without considering individual reliability, do not necessarily enhance evaluation reliability when unreliable judges are involved.
- We analyze the limitations of collective thought in the context of VLM evaluation and discuss potential strategies, such as selecting judges based on reliability metrics, to address them.

Our work offers insights for the design of evaluation frameworks, promoting the development of more reliable models. By addressing the challenges identified, we could pave the way for improved methodologies that can be used to effectively evaluate VLMs in handling real-world video content.

## 2 RELATED WORK

**Video Language Models** Video language models (Lin et al., 2023; Li et al., 2023c; Zhang et al., 2023) represent advanced models capable of handling a variety of video understanding tasks, including comprehension and captioning, as well as question-answering. These models process video and textual input to generate text-based outputs. Architecturally, VLMs typically integrate pre-trained vision backbones (Radford et al., 2021; Fang et al., 2023; Wang et al., 2022) with large language models (Touvron et al., 2023; Zheng et al., 2023) through connector modules such as MLP adapters, Q-former (Dai et al., 2023), and gated attention mechanisms (Alayrac et al., 2022). Early studies,

such as VideoChat (Li et al., 2023b) and VideoChat-GPT (Li et al., 2023c), used a two-stage training approach focused on alignment and adherence to video-related instructions. More advanced models have been developed recently, which enhance architectural frameworks (Li et al., 2023c), expand to new application areas (Munasinghe et al., 2023), and support longer videos (Song et al., 2023; Ren et al., 2023).

**Model Evaluation** VLMs are traditionally evaluated using metrics tailored to each specific task. For example, in image captioning, common metrics include BLEU (Papineni et al., 2002), METEOR (Banerjee & Lavie, 2005), ROUGE (Lin, 2004), and CIDER (Vedantam et al., 2015), which measure the similarity between generated captions and reference captions. Similarly, Visual Question Answering (VQA) tasks are evaluated using precision metrics that directly compare the responses of the models to those provided by human annotators (Agrawal et al., 2023; Mañas et al., 2023). However, these traditional metrics often fail to capture the nuanced details and subtleties in the responses produced by models, particularly in complex or subjective cases. In order to achieve a more comprehensive evaluation, human assessments are used to account for contextual and creative elements that automated metrics might overlook. However, the high costs make human evaluations unscalable. Recent studies have developed methods that leverage models to evaluate models. For example, numerous studies have utilized language models to assess the output of language models (Zhu et al., 2023; Li et al., 2023a; Kocmi & Federmann, 2023; Chiang & Lee, 2023). With the advancement of multimodal language models, recent studies have focused on using visual language models to evaluate responses from visual language models (Kim et al., 2023). Our study is the first systematic work to evaluate VLMs by using VLMs.

**Collective Decision-Making** Drawing from interdisciplinary theories, our methodology is based on the principles of collective intelligence and social psychology. The collective intelligence theory suggests that groups can achieve higher levels of intelligence and problem solving capability than isolated individuals (Malone et al., 2009). This is complemented by the "Wisdom of Crowds" principle, which argues that diverse groups can make better decisions than even the most capable individuals within them (Surowiecki, 2005). Furthermore, the social constructionist perspectives provide insight into how collective assessments evolve from the integration of multiple cognitive processes, reflecting broader sociocultural contexts (Burns & Engdahl, 1998). These theories inform our approach to synthesizing assessments from multiple VLMs into a comprehensive evaluation.

## 3    METHODOLOGY

In this section, we begin by elaborating the data collection process, where video-question pairs and their corresponding responses (generated by the VLM candidates, i.e., the VLMs to be evaluated) are gathered to serve as evaluation inputs for subsequent steps. Following this, we describe our approach for comparing individual reviews generated by VLMs with those produced through multi-LLM Agent-Debates, aiming to assess the reliability of each VLM. Finally, we present our proposed collective thought-based evaluation methodology for VLMs, which employs a structured three-stage process. Each stage is designed to evaluate the models' ability to interpret and respond to complex video content, thereby enhancing the reliability of VLMs as evaluative judges.

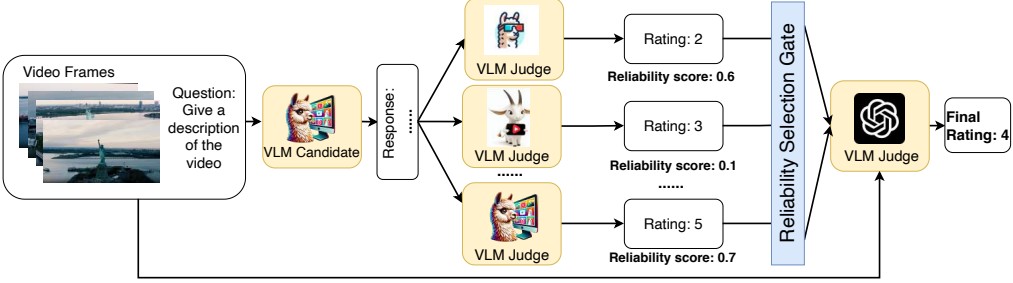

Figure 2: Diagram illustrating the multi-stage evaluation process involving multiple initial reviews, a reliability selection gate in the middle, and a final comprehensive review by an advanced model.

## 3.1 VLM Candidates Generate Responses

A video question pair is defined as $(v, t)$, where $v$ represents a video sequence, and $t$ is the corresponding textual instruction or query. There are two phases in data collection.

**Phase 1: Video-Question Pair Collection** The Complex Video Reasoning and Robustness Evaluation Suite (CVRR-ES) (Khattak et al., 2024) is a dataset that comprehensively assesses VLMs across 11 diverse real-world visual dimensions $V_d$ (see Table 2), such as interpretation of social context. From this dataset, we collect a set of video-question pairs $D = \{(v_1, t_1), (v_2, t_2), ..., (v_n, t_n)\}$, where each pair represents a unique video-instruction combination (see Figure 6 for an example).

**Phase 2: VLM Response Generation** For each pair of video questions $(v_i, t_i)$, we generate responses using a set of VLM candidates $M = \{M_1, M_2, ..., M_m\}$, which are to be evaluated. The response of model $M_j$ to pair $(v_i, t_i)$ is denoted as $r_{ij}$. The responses are then collected and cataloged, providing a rich source of data for subsequent analysis. We ensure that each response adheres to our predefined criteria for relevance and completeness, excluding any that do not meet the standards (see Figure 6 for an example).

## 3.2 Individual VLM Judge Reviews Candidate Responses

**Review by individual VLM** As shown in Figure 2, we employ a set of VLM judges $M^J$ to generate initial reviews $R = \{R_1, R_2, ..., R_q\}$ for each video-question-answer tuple $r_{ij}$ from the previous section. Each model offers a unique perspective, influenced by its training and inherent capability. These reviews are expected to offer various interpretations and assessments, reflecting the models' different ways to understand and analyze visual information (see Figure 6 for an example).

**Review by LLM Agents Debate (Reference-guided grading)** As shown in Figure 1, we engage a set of LLM agents to conduct discussions and generate initial reviews. The LLM agents receive both the responses generated by the VLM candidates and the reference responses provided by CVRR-ES (Khattak et al., 2024) for consideration. Through multiple rounds of interaction and debate, the LLM agents collaboratively refine their assessments. Finally, another LLM agent consolidates the insights and reaches a consensus on the rating (see Figure 6 for an example).

**Contrasting Reviews from LLM Agent-Debate and VLM** We assume that reviews generated through LLM Agent-Debate are the most reliable, as the inclusion of reference responses enhances the validity of the judgments made during these debates. To evaluate the agreement between LLM debates and VLM-generated reviews, we use *Weighted Cohen's Kappa* (Cohen, 1960; Artstein & Poesio, 2008), a statistical measure of the agreement between judges for categorical data. Unlike the unweighted version, which treats all disagreements equally, the weighted variant accounts for the extent of disagreement by assigning different weights to each category. This approach is particularly effective for ordinal categories, as it allows for partial credit in cases of minor disagreements:

$$\kappa = 1 - \frac{\sum_{\alpha,\beta} w_{\alpha\beta} O_{\alpha\beta}}{\sum_{\alpha,\beta} w_{\alpha\beta} E_{\alpha\beta}}$$

where $O_{\alpha\beta}$ is the observed frequency in which judge 1 assigned rating $\alpha$ and judge 2 assigned rating $\beta$, $E_{\alpha\beta}$ is the expected frequency for such assignments under the assumption of independent ratings, and $w_{\alpha\beta}$ is the weight assigned to the disagreement between categories $\alpha$ and $\beta$, which is typically calculated based on the squared or linear difference between categories. For the weighting function, we employ a quadratic weighting scheme defined as

$$w_{\alpha\beta} = 1 - \left(\frac{\alpha - \beta}{k - 1}\right)^2,$$

where $k$ represents the number of possible ratings, i.e., 5, and $\alpha$ and $\beta$ are integers between 1 and 5.

## 3.3 Collective VLM Judge Reviews Candidate Responses

The collective thought evaluation process is designed to harness the collective insights of multiple VLMs, followed by a comprehensive review using a more sophisticated model. This approach is

inspired by the concept of "wisdom of crowds" in collective intelligence, with the aim of leveraging the diverse strengths of various models to achieve a more accurate and nuanced assessment. By pooling insights from different models, we leverage a form of crowd-sourced thought to enhance the precision of decision-making when evaluating video content.

**Collective Thought Judge**  We utilize an advanced model $M_a^J$, which takes the video-question pair and response content as well as the corresponding reviews from VLM judges to generate a final assessment $A$:

$$A = M_a^J(r_{i,j}, R_1, R_2, \ldots, R_q)$$

Figure 2 shows the overall pipeline. After collecting the initial reviews, the advanced video language model aggregates these reviews along with the video-questions and responses to produce a consolidated final assessment. This model is designed to process and integrate multiple sources of information, enabling it to evaluate initial reviews and determine the most accurate and coherent response. Using all available information, the advanced judge provides a final review that could potentially reduce individual biases and improve the overall quality of the judgment. Notably, the advanced judge employed in this process is GPT-4o, which achieved the highest agreement with the LLM Agent-Debate.

**Mixture of Judges**  To further enhance the accuracy of the evaluation process, as shown in the reliability selection gate in Figure 2, we implement a *Mixture of Judges* strategy that leverages *Weighted Cohen's Kappa* to select the most reliable subset of VLMs $M^{J'} \subseteq M^J$ for each visual dimension $V_d$. Note that the *Weighted Cohen's Kappa* quantifies the agreement between model $M_e^J$ and the LLM Agent-Debate within the visual dimension $V_d$. Reliability scores $\kappa_{d,e}$ reflect the consistency with which each model aligns with the LLM debate within the given visual dimension.

For each visual dimension $V_d$, we select the subset $M^{J'}$ of models $M_e^J$ where $\kappa_{d,e}$ exceeds a predefined threshold $\theta$:

$$M^{J'} = \{M_e^J \mid \kappa_{d,e} \geq \theta\}$$

Alternatively, we may select the top $k$ models with the highest reliability scores for each visual dimension $V_d$:

$$M^{J'} = \{M_e^J \mid \kappa_{d,e} \text{ is among the top } k \text{ scores for visual dimension } V_d\}.$$

By dynamically selecting models based on their reliability scores at the visual dimension level, we ensure that only the most reliable and consistent models contribute to the final assessment.

## 4 EXPERIMENTAL SETUP

This section details the experimental setup used to evaluate the performance of VLMs as judges. We adopt an "Analyze-then-Judge" framework tailored to the video domain, where evaluators first examine the video content and then provide their assessments.

### 4.1 MODELS

| | |
|---|---|
| **Candidates** | Video-LLaVA, LLaMA-VID, GPT-4o mini, Video-ChatGPT, mPLUG-Owl-Video |
| **Judges (VLM)** | Video-LLaVA, LLaMA-VID, GPT-4o mini, InternVL2, GPT-4o |
| **Judges (LLM)** | GPT-3.5, Agents-Debate (GPT-4o text input only, GPT-3.5) |
| **Final Judge** | GPT-4o |

Table 1: List of candidate and judge models

Table 1 lists the candidate and judge models used in our study. During data collection, candidate models — Video-LLaVA, LLaMA-VID, GPT-4o mini, Video-ChatGPT and mPLUG-Owl-Video — are utilized to generate video-question pairs and their corresponding responses denoted as $r_{ij}$.

In the subsequent evaluation stage, we employ both VLMs and LLMs as judges. The GPT-3.5 model does not have access to the video content but is provided with reference answers. For LLM

agent debate, we use the GPT-3.5 and GPT-4o models in text-only mode (no visual input). They are also provided with reference answers and engage in debates. For VLM judging, we select the advanced models listed in the Judges (VLM) category in Table 1. We then compare the reviews obtained from individual VLM judges with those derived from the LLM or LLM debates to evaluate consistency and reliability. In the final stage, the advanced GPT-4o model reviews the video-question pairs and corresponding responses, together with the individual VLM judges' reviews, to produce a consolidated final assessment.

## 4.2 DATASET

The CVRR-ES (Khattak et al., 2024) dataset encompasses a variety of visual dimensions $V_d$ that cover diverse video categories relevant to real-world scenarios. These visual dimensions range from context-dependent areas, such as social and emotional contexts, to common video types, including unusual activities (Khattak et al., 2024). The dataset comprises 2,400 high-quality open-ended question-answer (QA) pairs derived from 217 meticulously curated videos. The videos have an average duration of 22.3 seconds, with lengths varying from a minimum of 2 seconds to a maximum of 183 seconds (Khattak et al., 2024). Some samples of this dataset are listed in Table 3.

## 4.3 EVALUATION CRITERIA AND METRICS

A scoring-based evaluation approach is used to evaluate VLM candidates. Each evaluation (i.e., review) for a video-question pair and its corresponding response is a score that reflects its accuracy and relevance. In our experiments, the VLM judges are required to provide a rating on a scale from 1 to 5, where 1 represents the poorest performance and 5 indicates perfect accuracy. For additional metrics, such as pair comparison, we simply use the scores from the Scoring Evaluation. For example, when making a pairwise comparison between two VLM candidates, the candidate with a higher rating score is deemed superior.

# 5 EXPERIMENTAL RESULTS

## 5.1 INDIVIDUAL VLM JUDGE REVIEWS CANDIDATE RESPONSES

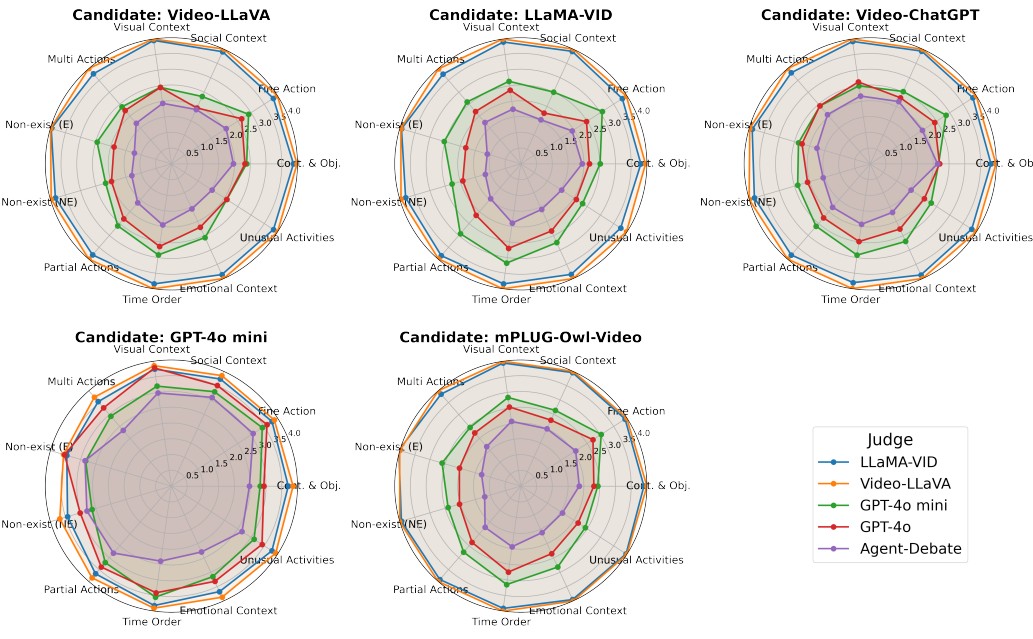

Figure 3: Scores assigned by judges to candidates on various visual dimensions.

Figure 3 displays the evaluation scores assigned by various judges to the candidate VLMs across different visual dimensions (see Table 2). Additional details are provided in Table 4. By analyzing the results, we observe that some VLMs consistently assign high scores to candidate models across all visual dimensions. For instance, when judged by Video-LLaVA, the candidate models receive scores close to 4.00 across nearly all dimensions. Similarly, LLaMA-VID as a judge also assigns high scores, typically higher than 3.70. This trend indicates that some VLMs may have a tendency to give overly positive evaluations to candidate models. In contrast, the LLM judges and the Agent-Debate method assign significantly lower scores. The Agent-Debate method, which involves multiple LLM agents participating in the discussion (with reference responses) and reaching a consensus, consistently gives the lowest scores among all judges. For example, in the case of LLaMA-VID as a candidate model, the Agent-Debate scores range from 1.46 to 1.94 across different dimensions. This suggests that LLM agents provide a more critical assessment than VLM judges. Some output samples from various VLM judges can be found in Appendix A.

The disparity between VLM and LLM judges highlights potential reliability issues with VLMs as judges. VLMs may be prone to overestimating the performance of candidate models due to their limited ability to fully understand the content, or due to inherent biases. This overestimation can lead to inflated scores that do not accurately reflect the true capabilities of the candidate models. However, the Agent-Debate method appears to provide a more stringent and possibly more accurate evaluation. By engaging multiple LLM agents in a debate with the provided reference responses and reaching a consensus, this method reduces individual biases. However, reference responses are required for the LLMs.

Additionally, certain visual dimensions consistently receive lower scores across all judges. For example, the dimensions of Non-exist (E) and Non-exist (NE) often have lower scores, indicating that candidate models struggle to detect non-existent entities or events in video content. This highlights specific areas where VLMs need improvement to handle video understanding tasks effectively.

Among the various VLMs, GPT-4o and the Agent-Debate method have the most similar evaluation patterns. They consistently assign lower scores to candidate models in most visual dimensions, reflecting a more stringent and critical assessment. For example, when evaluating the candidate model LLaMA-VID, GPT-4o assigns scores ranging from 1.77 to 2.48, and Agent-Debate assigns scores from 1.11 to 1.94. This similarity suggests that GPT-4o and the Agent-Debate method exhibit similar levels of rigor in assessing the performance of candidate models, potentially offering more reliable and unbiased evaluations compared to other judge models.

In addition, the left chart in Figure 4 illustrates the rating statistics of all judges. Video-LLaVA and LLaMA-VID show a significant concentration of ratings at 4. In contrast, GPT-4o and Agent-Debate have higher counts with lower ratings. InternVL2 and GPT-4o mini display a more balanced distribution across ratings.

## 5.2 Contrasting Reviews from VLM and LLM Agent-Debate

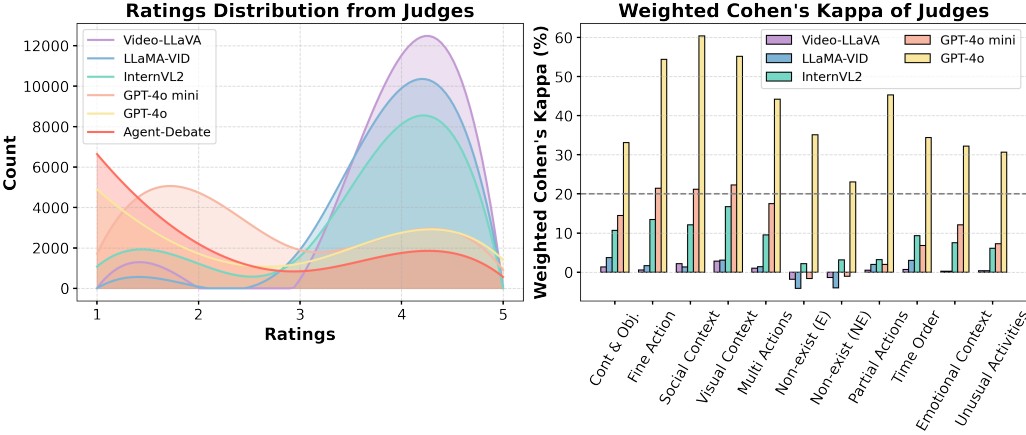

Figure 4: Left: Distribution of ratings. Right: Judges' weighted Cohen's Kappa values.

The right chart in Figure 4 illustrates the average agreement scores, expressed as percentages, between various VLMs and the Agent-Debate method across visual dimensions. The agreement is quantified using the *Weighted Cohen's Kappa* (Cohen, 1960; Artstein & Poesio, 2008), where higher values indicate a greater agreement. It is generally accepted that values less than or close to 0 indicate no agreement, while values between 0 and 0.20 are considered to represent slight agreement, 0.21–0.40 as fair agreement, 0.41–0.60 as moderate agreement, 0.61–0.80 as substantial agreement, and 0.81–1.00 as indicating almost perfect agreement.

From the chart, we observe that VLMs such as Video-LLaVA and LLaMA-VID have relatively low agreement scores with the Agent-Debate method, often including negative values. For example, Video-LLaVA shows agreement scores ranging from $-1.83$ to $2.81$, and LLaMA-VID ranges from $-4.15$ to $3.70$. In contrast, InternVL2, GPT-4o mini, and particularly GPT-4o exhibit significantly higher agreement scores, indicating substantial agreement with the Agent-Debate evaluations. GPT-4o, for example, shows agreement scores exceeding $50$ in several dimensions, such as $60.38$ for Social Context and $55.18$ for Visual Context. More details are elaborated in Table 5.

The results suggest that GPT-4o is more aligned with the Agent-Debate evaluations, potentially due to its multimodal capabilities allowing better critical analysis. However, the significant disagreement for other VLMs, such as Video-LLaMA, raises concerns about the reliability of using them as judges, possibly due to the inability to fully understand the content. The Agent-Debate method with reference responses provided, involving multiple LLM agents engaging in discussion and reaching a consensus, provides a more reliable evaluation by mitigating individual misjudgment. The collaborative nature of the Agent-Debate method reduces the impact of any single agent's misjudgment.

## 5.3 COLLECTIVE VLM JUDGE REVIEWS CANDIDATE RESPONSES

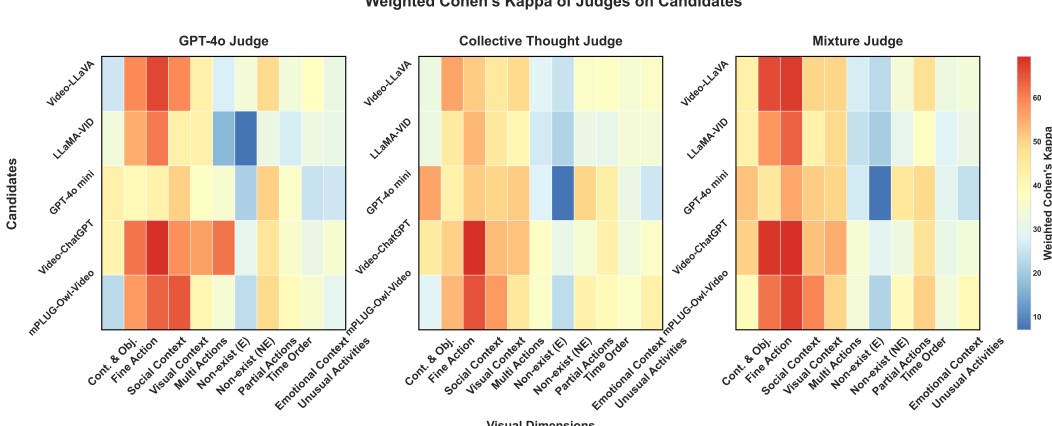

Figure 5: Left: Weighted Cohen's Kappa for the GPT-4o judge across various candidate models; Middle: Weighted Cohen's Kappa for the Collective Thought judge across various candidate models; Right: Weighted Cohen's Kappa for the Mixture judge across various candidate models.

**Collective Thought Judge** The middle chart in Figure 5 and Table 6 present the agreement scores between the collective evaluations of all judges and the Agent-Debate method in various visual dimensions. The judges encompass both reliable models (with high agreement scores) and models with known reliability issues (with low agreement scores). The initial judges include LLaMA-VID, Video-ChatGPT, Video-LLaVA, and GPT-4o mini. The final judge is GPT-4o.

From the results, it is evident that including both more reliable and less reliable judges in the collective review process does not necessarily improve the reliability of the ratings. The average agreement scores with the Agent-Debate method remain moderate to low, and in some cases the agreement is lower than when using only GPT-4o as the judge. For example, in collective evaluation, the average agreement scores in dimensions range from $2.72$ to $42.53$, with the highest scores observed in dimensions such as Social Context and Visual Context, as shown in Table 6. These scores are lower than those scores achieved by individual GPT-4o judges. Inclusion of less reliable judges introduces

noise and biases in collective assessment. As a result, the collective ratings do not align closely with the baseline established by the Agent-Debate method. This phenomenon highlights the challenges of aggregating evaluations from heterogeneous judges without proper weighting or selection mechanisms.

These findings indicate that unreliable models can adversely affect the results of the ensemble methods. In the context of VLM evaluation, where hallucinations and biases are prevalent (Tong et al., 2024), the negative impact is pronounced.

**Mixture of Judges** The right chart in Figure 5 illustrates the agreement scores when employing a mixture of judges, selected based on their per-dimension reliability scores (*Weighted Cohen's Kappa*). Additional details are provided in Table 7. The selection of judges is dynamic, as outlined in section 3.3. For example, for the fine action visual dimension, the selected judges include InternVL2 and GPT-4o mini, with GPT-4o serving as the final judge. The goal of this strategy is to enhance the reliability of the evaluation by including only the most reliable judges for each visual dimension.

Despite this selective approach, the results indicate that the mixture of judges does not substantially improve the agreement with the Agent-Debate method. The average agreement scores are comparable to those observed in the collective thought approach with all judges. For example, the average agreement scores range from 2.72% to 60.70% across different dimensions. Even though the judges are selected for higher reliability in specific categories, overall improvement in evaluation accuracy is still lower than the performance achieved by using GPT-4o alone.

One reason could be that the reliability scores used for the selection of judges may not fully capture the judges' ability to evaluate complex video content. As a result, the selected judges might still exhibit biases or hallucinations. These findings suggest that simply selecting judges based on previous performance metrics does not guarantee improved evaluation outcomes. The intricacies of multimodal evaluation require more advanced methods that can effectively integrate judgments while mitigating the misjudgment of the individual model.

## 6 DISCUSSION

**Reliability of Individual VLMs as Judges** Our results indicate that VLMs, including Video-LLaVA, consistently overestimate the scores of candidate models. This overestimation may result from their inability to fully comprehend the content or from inherent biases in training data. For example, if the data are representative of more positive feedback, then the model would be naturally prone to giving higher ratings. GPT-4o is the only VLM that exhibits significant agreement with the Agent-Debate method and can be considered more reliable as a judge.

We compared the judges' *Weighted Cohen's Kappa* scores with the performance scores from (Khattak et al., 2024) and observed a consistent trend: the better the judges perform on the benchmark, the higher are their *Weighted Cohen's Kappa* scores. This finding suggests that a judge can be reliable only if it demonstrates a strong understanding of the content itself. To improve reliability, we fine-tune the underperforming VLM model Video-LLaVA. As shown in Figure 7, despite being fine-tuned, Video-LLaVA's rating distribution remains skewed toward higher ratings, and its *Weighted Cohen's Kappa* scores, i.e., reliability as a judge, are only slightly improved. The agreement scores with the benchmark do not approach those of GPT-4o. These findings indicate that simply improving a model's comprehension ability is insufficient to enhance its reliability as a judge.

**Weak to Strong Evaluation** The results in Table 4 and Table 5 indicate that the weaker VLMs, such as Video-LLaVA, judging stronger models, such as GPT-4o mini, result in unreliable evaluations since these weaker models lack the necessary understanding and critical reasoning abilities. This is in accordance with recent research on weak-to-strong generalization in language models, which shows that if one naively fine-tunes strong models with labels from weaker supervisors, not all of the capabilities of the stronger models are being tapped into (Burns et al., 2023). Equally important is our finding that **much stronger models cannot be reliably evaluated by weaker VLMs alone**. This calls for the development of more sophisticated methods for evaluation, to ensure reliable alignment and performance in such advanced VLMs.

**Limitations of Collective Thought Approaches** Our experiments with collective thought approaches do not yield significant improvements in evaluation reliability. The mixing of reliable and unreliable judges introduces noise. Even when selecting judges based on reliability scores, the mixture of judges does not substantially enhance agreement with the Agent-Debate method. Notably, we find that GPT-4o, when used as a sole judge, outperforms itself when paired with a group of less reliable judges. This indicates that GPT-4o is affected by the presence of incorrect opinions within the collective, highlighting its vulnerability to noise introduced by less reliable judges.

**Extension to Another Dataset.** To verify the generality of our findings, we also performed experiments on the VideoChatGPT dataset[1]. The results are presented in Appendix F. They are consistent with those from the CVRR-ES: less capable VLMs (e.g., Video-LLaVA) systematically over-rate candidates, whereas GPT-4o maintains moderate agreement with the text-only reference-guided judge. These additional experiments reinforce our conclusion that less capable VLMs are unreliable as judges in different datasets and conditions.

**Implications and Future Work** The results underscore the importance of employing reliable and robust evaluation frameworks for VLMs. Relying solely on individual VLMs for evaluation is inadequate due to their limited content understanding and lack of critical analysis. The Agent-Debate method, leveraging collaborative reasoning among multiple agents, provides a more accurate assessment of VLM performance. In future work, we will study the effectiveness of an iterative collective thought approach, exploring how multi-round discussions among VLM agents can further enhance evaluation reliability and mitigate the limitations observed with the current aggregation method.

## 7 Conclusion and Limitations

In this paper, we conduct a comprehensive evaluation of Video Language Models using a multistage methodology that includes individual assessments by VLMs and a collaborative Agent-Debate approach. Our study aims to determine the most reliable model for evaluating VLMs in complex video understanding tasks.

Our findings highlight several key insights:

- **Reliability of VLMs as Judges:** Some VLM judges tend to overestimate the performance of candidate VLMs, likely due to limited content understanding or inherent biases. GPT-4o is the only model that exhibits significant reliability as judge.

- **Towards Improving VLM Reliability as Judge:** A reliable judge must possess not only strong comprehension skills but also advanced abilities in evaluation and critical analysis. To improve reliability, we can incorporate specialized training to enhance both understanding and evaluation skills, and reduce biases and hallucinations.

- **Limitations of Collective Thought Approaches:** Collective evaluation methods that aggregate judgments from both reliable and unreliable models do not necessarily enhance evaluation accuracy. The inclusion of less reliable judges introduces noise and biases, diminishing the overall reliability of the assessment. Even when a mixture of judges selected based on reliability scores is employed, no significant improvements are observed.

Although our study provides valuable insights, several limitations should be acknowledged. Dependence on Specific Datasets and Models: Due to limited resources, our experiments are based on the two datasets and a selected set of VLMs and LLMs. The generalizability of our findings to other datasets and models may be limited. Scope of Evaluation Methods: We focused on the Agent-Debate method and collective thought approaches involving VLMs and LLMs. Other evaluation strategies, such as human expert assessments or alternative ensemble methods, are not explored in this study. Quantitative Metrics: The reliance on agreement scores based on *Weighted Cohen's Kappa* provides a quantitative measure of agreement, but may not fully capture the qualitative aspects of model evaluation. Subtle nuances in judgments might not be reflected in these metrics. Computational Resources and Costs: The use of advanced models such as GPT-4o incurs significant computational costs. This may limit the practicality of deploying such evaluation methods at scale or in resource-constrained environments.

---

[1] https://huggingface.co/datasets/lmms-lab/VideoChatGPT

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

APPENDIX

- Qualitative Samples
- Dataset Details
- Details on Ratings
- Details on Agreement
- Compare GPT-4o, collective thought, mixture of judge
- Extension to the VideoChatGPT Dataset
- Ablation Study on finetuning Video-LLaVA

## A  QUALITATIVE SAMPLES

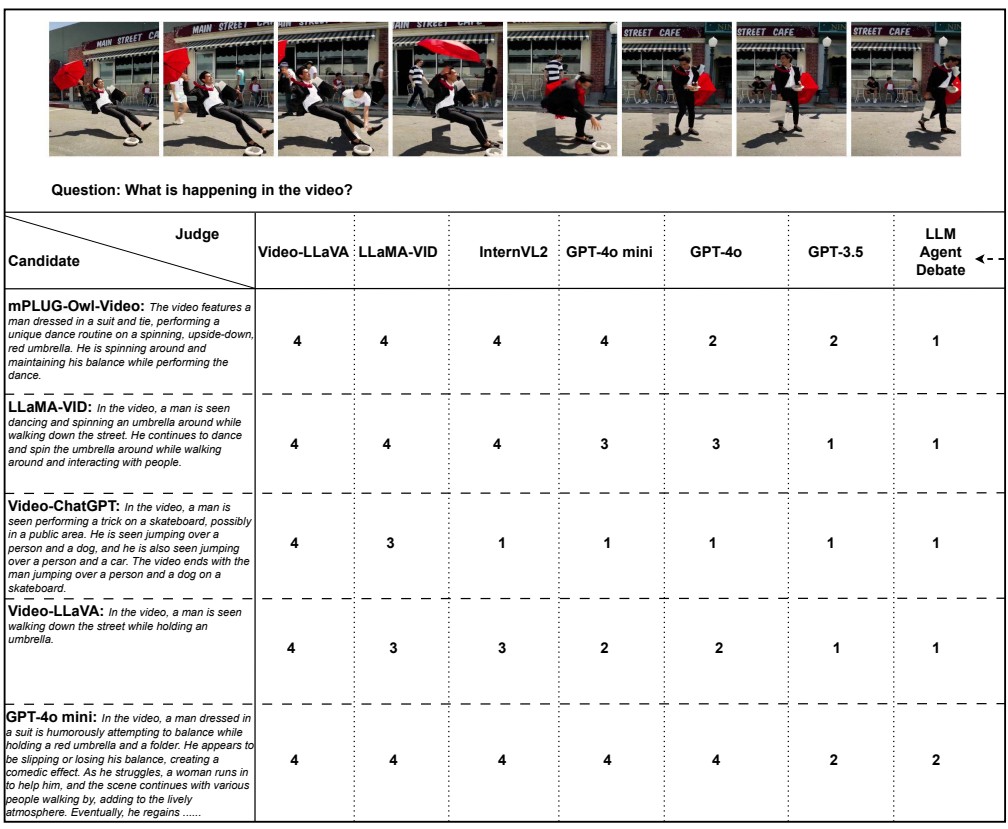

Figure 6: An example from the CVRR dataset, with candidate responses and corresponding ratings from the judges. Due to space constraints, only the rating numbers are displayed. In our setup, each judge provides both reasoning and a rating number. We have limited the selection of judges to the models listed, as other models demonstrate poor instruction-following abilities and tend to only provide a rating number.

## B  DATASET DETAILS

| Visual Dimension | Definition |
|---|---|
| 1) Multiple actions in a single video **(Multi Actions)** | This category includes videos with 2-4 activities, mostly featuring humans performing multiple actions. It tests the model's ability to reason about and understand interrelations between different actions. |
| 2) Fine-grained action understanding **(Fine Action)** | Focuses on subtle human activities like pushing, opening, closing, etc. Challenges the model's comprehension of fine-grained actions through carefully crafted questions. |
| 3) Partial actions **(Partial Actions)** | Features videos with actions likely to be followed by subsequent actions, but not executed. Tests the model's ability to avoid generating contextually relevant but non-existent content. |
| 4) Time order understanding **(Time Order)** | Assesses the model's ability to recognize temporal sequences of activities, crucial for distinguishing between atomic actions like pushing and pulling. |
| 5) Non-existent actions with existent scene depictions **(Non-exist (E))** | Examines the model's robustness in scenarios with introduced non-existent activities without altering the physical and spatial scenes. |
| 6) Non-existent actions with non-existent scene depictions **(Non-exist (NE))** | Evaluates the model's reliability in handling questions with both non-existent activities and scene comprehension, testing its ability to avoid generating imaginary content. |
| 7) Continuity and object instance count **(Cont. & Obj.)** | Tests the model's ability to accurately recognize and count object instances, and distinguish between existing and newly introduced objects in a scene. |
| 8) Unusual and physically anomalous activities **(Unusual Activities)** | Assesses the model's ability to understand unconventional activities that seem to defy physics, testing generalization to out-of-distribution scenarios. |
| 9) Interpretation of social context **(Social Context)** | Evaluates the model's ability to infer the rationale behind actions based on social context, using diverse videos with challenging questions. |
| 10) Understanding of emotional context **(Emotional Context)** | Tests the model's capacity to interpret actions considering emotional context, using videos and questions focused on recognizing action nature based solely on emotional cues. |
| 11) Interpretation of visual context **(Visual Context)** | Focuses on the model's ability to recognize actions using overall visual contextual cues, requiring reasoning based on visual elements like shadows. |

Table 2: Definitions of Visual Dimensions for Video Understanding (Khattak et al., 2024).

Table 3: Examples of question-answer pairs in the CVRR-ES benchmark for various complex video evaluation dimensions. The content is collected from previous work (Khattak et al., 2024).

| Visual Dimension | Sample Question-Answer Pairs |
|---|---|
| 1. Multiple actions in a single video | **Q:** Does the person stand up to welcome the cat or remain seated? |
| | **A:** The person remains seated throughout their interaction with the cat. 
 **Q:** What is the next action after using the laptop? 
 **A:** Placing a bag in the refrigerator. |
| 2. Fine-grained action understanding | **Q:** Does the man use the thread to sew fabric? |
| | **A:** No, he uses it to create loops and demonstrate tying a knot. 
 **Q:** What action is performed by the person's hands? 
 **A:** Plugging a black USB charging cable into the charging port. |
| 3. Partial actions | **Q:** What is happening in the video? 
 **A:** The video shows a red car door and a hand reaching for the handle. 
 **Q:** Is the snack replaced to its original position? 
 **A:** No, the video only shows moving the snack from right to left. |
| 4. Time order understanding | **Q:** Is liquid being taken out of the soda can? 
 **A:** No, the video doesn't show this activity. 
 **Q:** In which direction is the person running on the track? 
 **A:** The person is running anticlockwise. |
| 5. Non-existent actions with existent scene depictions | **Q:** Does the person clean around the sink after going through the bag? |
| | **A:** No, the person does not clean the area around the sink. 
 **Q:** How does the audience react to the keynote speaker? 
 **A:** The scene does not include a keynote speaker delivering a speech. |
| 6. Non-existent actions with non-existent scene depictions | **Q:** How do children interact with the flowers? |
| | **A:** There are no children or flowers depicted in the video. 
 **Q:** How does the child react when the dog runs past? 
 **A:** There is no child or dog in the video. |
| 7. Continuity and Object Instance Count | **Q:** How many unique sunglasses appear in the video? |
| | **A:** There are 4 unique sunglasses, one for each person in the car. 
 **Q:** Did the men's attire change when they re-entered the frame? 
 **A:** Yes, their attire changed upon re-entering the frame. |
| 8. Unusual and Physically Anomalous activities | **Q:** How does the person reach an elevated position? |
| | **A:** They ascended and floated in the air, not by walking or running. 
 **Q:** How is the person able to fly over the water? 
 **A:** They are using a flyboard system attached to their shoes. |
| 9. Interpretation of social context | **Q:** How did the crowd respond to the girl landing the water bottle? |
| | **A:** The crowd applauded to show appreciation for her success. 
 **Q:** Why does the boy touch ashes before touching the goat? 
 **A:** He uses the ashes to warm the goat, showing care. |
| 10. Understanding of emotional context | **Q:** Is the emotional context of the video negative? |
| | **A:** No, it is overwhelmingly positive. 
 **Q:** What is the nature of the interaction between the two individuals? 
 **A:** The interaction is friendly, evidenced by a warm hug and handshake. |
| 11. Interpretation of visual context | **Q:** Does the person undergo a real physical transformation? |
| | **A:** No, they remove a rubber mask, revealing they are a woman. 
 **Q:** What unusual behavior is shown between a predator and prey? 
 **A:** A cat plays and sleeps with chicks instead of hunting them. |

# C  DETAILS ON RATINGS

| Candidate | Visual Dimensions | | | | | | | | | | |
|---|---|---|---|---|---|---|---|---|---|---|---|
| **& Judge** | Cont. & Obj. | Fine Action | Social Context | Visual Context | Multi Actions | Non-exist (E) | Non-exist (NE) | Partial Actions | Time Order | Emotional Context | Unusual Activities |
| **LLaMA-VID** | | | | | | | | | | | |
| Video-LLaVA | 3.95 | 3.98 | 4.00 | 4.00 | 3.98 | 3.96 | 3.96 | 3.99 | 4.00 | 4.00 | 3.98 |
| LLaMA-VID | 3.81 | 3.83 | 3.92 | 3.90 | 3.76 | 3.94 | 3.81 | 3.85 | 3.84 | 3.85 | 3.76 |
| GPT-4o mini | 2.51 | 3.07 | 2.50 | 2.64 | 2.60 | 2.53 | 2.27 | 2.93 | 3.18 | 2.74 | 2.33 |
| InternVL2 | 3.12 | 3.63 | 3.31 | 3.36 | 3.55 | 3.46 | 3.12 | 3.63 | 3.56 | 3.34 | 3.09 |
| GPT-4o | 2.18 | 2.48 | 1.77 | 2.36 | 2.20 | 1.82 | 1.92 | 2.16 | 2.70 | 2.34 | 2.09 |
| GPT-3.5 | 2.31 | 2.30 | 1.85 | 2.05 | 2.01 | 1.06 | 1.17 | 1.73 | 2.24 | 1.90 | 1.95 |
| Agent-Debate | 1.94 | 1.93 | 1.53 | 1.75 | 1.73 | 1.11 | 1.17 | 1.46 | 1.89 | 1.58 | 1.54 |
| **GPT-4o mini** | | | | | | | | | | | |
| Video-LLaVA | 3.86 | 3.88 | 3.85 | 3.86 | 3.73 | 3.57 | 3.69 | 3.84 | 3.91 | 3.87 | 3.93 |
| LLaMA-VID | 3.71 | 3.79 | 3.73 | 3.75 | 3.54 | 3.46 | 3.43 | 3.67 | 3.82 | 3.67 | 3.79 |
| GPT-4o mini | 2.81 | 3.43 | 3.29 | 3.20 | 2.93 | 2.83 | 2.63 | 3.20 | 3.55 | 3.15 | 3.12 |
| InternVL2 | 3.14 | 3.59 | 3.57 | 3.57 | 3.39 | 3.33 | 3.20 | 3.43 | 3.60 | 3.47 | 3.64 |
| GPT-4o | 2.93 | 3.60 | 3.51 | 3.78 | 3.28 | 3.53 | 3.01 | 3.39 | 3.41 | 3.32 | 3.42 |
| GPT-3.5 | 3.41 | 3.90 | 3.95 | 3.73 | 3.21 | 3.80 | 3.74 | 3.73 | 3.93 | 3.20 | 3.49 |
| Agent-Debate | 2.47 | 3.08 | 3.09 | 2.98 | 2.34 | 2.86 | 2.79 | 2.81 | 2.40 | 2.29 | 2.66 |
| **Video-ChatGPT** | | | | | | | | | | | |
| Video-LLaVA | 3.99 | 4.00 | 3.99 | 4.00 | 4.00 | 3.98 | 4.00 | 3.99 | 3.99 | 3.99 | 3.99 |
| LLaMA-VID | 3.83 | 3.87 | 3.93 | 3.92 | 3.82 | 3.90 | 3.83 | 3.82 | 3.80 | 3.87 | 3.83 |
| GPT-4o mini | 2.17 | 2.86 | 2.52 | 2.50 | 2.43 | 2.37 | 2.40 | 2.62 | 2.92 | 2.70 | 2.29 |
| InternVL2 | 3.02 | 3.42 | 3.25 | 3.21 | 3.37 | 3.33 | 3.03 | 3.41 | 3.53 | 3.21 | 3.07 |
| GPT-4o | 2.20 | 2.43 | 2.30 | 2.62 | 2.44 | 2.25 | 2.07 | 2.26 | 2.49 | 2.27 | 2.04 |
| GPT-3.5 | 2.66 | 2.34 | 2.60 | 2.53 | 2.48 | 1.75 | 1.62 | 2.17 | 2.34 | 2.05 | 1.96 |
| Agent-Debate | 2.14 | 1.97 | 2.17 | 2.17 | 2.07 | 1.76 | 1.56 | 1.82 | 1.93 | 1.69 | 1.54 |
| **mPLUG-Owl-Video** | | | | | | | | | | | |
| Video-LLaVA | 4.00 | 3.99 | 4.00 | 2.07 | 4.00 | 4.00 | 4.00 | 4.00 | 4.00 | 4.00 | 4.00 |
| LLaMA-VID | 3.90 | 3.93 | 3.95 | 3.94 | 3.86 | 4.03 | 3.94 | 3.92 | 3.91 | 3.95 | 3.97 |
| GPT-4o mini | 2.44 | 3.03 | 2.64 | 2.84 | 2.46 | 2.60 | 2.40 | 2.77 | 3.15 | 2.82 | 2.43 |
| InternVL2 | 3.22 | 3.64 | 3.44 | 3.49 | 3.49 | 3.53 | 3.16 | 3.54 | 3.70 | 3.44 | 3.46 |
| GPT-4o | 2.32 | 2.72 | 2.53 | 2.54 | 2.23 | 2.03 | 2.03 | 2.36 | 2.75 | 2.35 | 2.16 |
| GPT-3.5 | 2.27 | 2.46 | 2.35 | 2.34 | 2.02 | 1.11 | 1.19 | 1.73 | 2.27 | 1.95 | 1.57 |
| Agent-Debate | 1.85 | 2.07 | 2.00 | 2.07 | 1.65 | 1.30 | 1.19 | 1.73 | 1.94 | 1.62 | 1.57 |
| **Video-LLaVA** | | | | | | | | | | | |
| Video-LLaVA | 3.98 | 3.99 | 4.00 | 4.00 | 4.00 | 3.99 | 3.97 | 4.00 | 4.00 | 3.99 | 4.00 |
| LLaMA-VID | 3.87 | 3.84 | 3.91 | 3.95 | 3.78 | 3.97 | 3.83 | 3.82 | 3.84 | 3.86 | 3.85 |
| GPT-4o mini | 2.38 | 2.91 | 2.35 | 2.45 | 2.39 | 2.46 | 2.17 | 2.60 | 2.91 | 2.57 | 2.09 |
| InternVL2 | 3.28 | 3.65 | 3.39 | 3.36 | 3.43 | 3.38 | 3.17 | 3.67 | 3.59 | 3.29 | 3.10 |
| GPT-4o | 2.33 | 2.65 | 1.95 | 2.45 | 2.24 | 2.21 | 2.08 | 2.26 | 2.64 | 2.21 | 2.08 |
| GPT-3.5 | 2.31 | 2.30 | 1.85 | 2.05 | 2.01 | 1.06 | 1.17 | 1.73 | 2.24 | 1.90 | 1.95 |
| Agent-Debate | 1.97 | 2.07 | 1.89 | 1.94 | 1.70 | 1.22 | 1.31 | 1.63 | 1.95 | 1.57 | 1.53 |

Table 4: Scores of candidate models given by judge models across various visual dimensions.

# D  DETAILS ON AGREEMENT

| Judge & Candidate | Cont. & Obj. | Fine Action | Social Context | Visual Context | Multi Actions | Non-exist (E) | Non-exist (NE) | Partial Actions | Time Order | Emotional Context | Unusual Activities |
|---|---|---|---|---|---|---|---|---|---|---|---|
| **Video-LLaVA** | | | | | | | | | | | |
| Video-LLaVA | -0.02 | 0.64 | 0.00 | 0.00 | 0.00 | 0.04 | -0.13 | 0.09 | 0.00 | 0.01 | 0.00 |
| LLaMA-VID | 0.93 | -0.35 | 0.00 | -0.03 | 0.07 | 0.09 | -1.02 | -0.61 | 0.00 | -0.14 | 0.09 |
| GPT-4o mini | 5.34 | 2.01 | 10.69 | 14.09 | 4.91 | -8.05 | -5.77 | 3.20 | 3.41 | 1.53 | 2.18 |
| Video-ChatGPT | 0.48 | 0.14 | 0.05 | 0.00 | 0.00 | -1.24 | 0.00 | -0.32 | -0.01 | -0.17 | -0.44 |
| mPLUG-Owl-Video | 0.00 | 0.18 | 0.00 | 0.00 | 0.00 | 0.00 | 0.00 | 0.00 | 0.00 | -0.06 | 0.00 |
| *Average* | 1.35 | 0.52 | 2.15 | 2.81 | 1.00 | -1.83 | -1.38 | 0.47 | 0.68 | 0.23 | 0.37 |
| **LLaMA-VID** | | | | | | | | | | | |
| Video-LLaVA | 0.66 | 1.42 | -1.42 | 0.27 | 0.11 | -0.55 | -0.67 | 0.59 | 2.64 | -0.12 | 0.39 |
| LLaMA-VID | 2.26 | -0.27 | -0.03 | -1.16 | -1.11 | -0.02 | -2.63 | -0.81 | 2.31 | -0.03 | 0.21 |
| GPT-4o mini | 10.62 | 9.06 | 7.46 | 16.13 | 5.64 | -17.09 | -14.94 | 9.04 | 5.99 | 3.88 | -0.53 |
| Video-ChatGPT | 3.72 | -0.01 | 0.62 | 0.40 | 3.58 | -3.42 | -1.91 | 0.38 | 1.68 | -0.65 | -1.21 |
| mPLUG-Owl-Video | 1.24 | -0.15 | -1.41 | -0.25 | -1.28 | 0.32 | -0.08 | 0.81 | 2.31 | -0.76 | -0.09 |
| *Average* | 3.70 | 2.01 | 1.04 | 3.08 | 1.39 | -4.15 | -4.04 | 2.00 | 2.99 | 0.23 | 0.37 |
| **GPT-4o mini** | | | | | | | | | | | |
| Video-LLaVA | 11.72 | 23.37 | 19.88 | 25.00 | 18.66 | 4.39 | -1.96 | -4.02 | 5.75 | 17.65 | 15.14 |
| LLaMA-VID | 11.72 | 23.37 | 19.88 | 25.00 | 18.66 | 4.39 | -1.96 | -4.02 | 5.75 | 17.65 | 15.14 |
| GPT-4o mini | 28.59 | 19.43 | 23.18 | 20.10 | 26.19 | -5.55 | -28.86 | 16.54 | 6.85 | 3.08 | -8.32 |
| Video-ChatGPT | 10.58 | 18.04 | 21.01 | 17.99 | 9.20 | -15.19 | -18.65 | -7.23 | 0.68 | 5.91 | 6.75 |
| mPLUG-Owl-Video | 9.87 | 22.94 | 21.96 | 23.31 | 14.94 | 3.84 | -3.31 | -1.56 | 14.86 | 16.27 | 7.66 |
| *Average* | 14.50 | 21.43 | 21.18 | 22.28 | 17.53 | -1.62 | -1.09 | 1.94 | 6.78 | 12.11 | 7.27 |
| **InternVL2** | | | | | | | | | | | |
| Video-LLaVA | 6.57 | 8.48 | 6.74 | 10.23 | 4.30 | 1.09 | 0.52 | 2.99 | 2.60 | 6.94 | 5.90 |
| LLaMA-VID | 22.68 | 22.34 | 22.44 | 32.12 | 20.03 | 5.65 | 10.46 | 15.93 | 19.29 | 11.86 | 8.40 |
| GPT-4o mini | 8.75 | 14.78 | 9.82 | 16.79 | 11.80 | -1.98 | -0.63 | -1.78 | 6.57 | 3.06 | 6.43 |
| Video-ChatGPT | 8.75 | 14.78 | 9.82 | 16.79 | 11.80 | -1.98 | -0.63 | -1.78 | 6.57 | 3.06 | 6.43 |
| mPLUG-Owl-Video | 6.15 | 9.43 | 11.24 | 13.73 | 6.36 | 2.89 | 0.63 | -1.26 | 8.29 | 7.24 | 2.97 |
| Video-LLaVA | 9.31 | 12.30 | 10.15 | 10.70 | 5.11 | 3.24 | 4.52 | 0.20 | 9.89 | 8.40 | 6.78 |
| *Average* | 10.69 | 13.47 | 12.08 | 16.71 | 9.52 | 2.18 | 3.10 | 3.21 | 9.33 | 7.50 | 6.10 |
| **GPT-4o** | | | | | | | | | | | |
| Video-LLaVA | 25.16 | 58.85 | 66.35 | 59.28 | 43.14 | 27.01 | 34.00 | 48.85 | 33.57 | 37.80 | 31.75 |
| LLaMA-VID | 33.66 | 54.43 | 60.79 | 43.13 | 43.08 | 16.74 | 6.99 | 31.50 | 26.41 | 32.23 | 30.87 |
| GPT-4o mini | 42.58 | 39.71 | 42.07 | 50.45 | 37.04 | 35.67 | 21.15 | 50.29 | 36.87 | 24.59 | 25.22 |
| Video-ChatGPT | 41.44 | 61.71 | 69.39 | 58.41 | 56.87 | 61.53 | 29.74 | 46.53 | 36.29 | 31.45 | 35.47 |
| mPLUG-Owl-Video | 22.73 | 57.36 | 63.28 | 64.64 | 41.08 | 34.49 | 23.56 | 49.32 | 38.89 | 35.09 | 29.93 |
| *Average* | 33.11 | 54.41 | 60.38 | 55.18 | 44.24 | 35.09 | 23.09 | 45.30 | 34.40 | 32.23 | 30.65 |

Table 5: Agreement scores across various visual dimensions between VLMs and LLM Agent-Debate

## E   COMPARE GPT-4O, COLLECTIVE THOUGHT, MIXTURE OF JUDGES

| Judge | visual dimensions | | | | | | | | | | |
|---|---|---|---|---|---|---|---|---|---|---|---|
| **& Candidate** | *Cont. & Obj.* | *Fine Action* | *Social Context* | *Visual Context* | *Multi Actions* | *Non-exist (E)* | *Non-exist (NE)* | *Partial Actions* | *Time Order* | *Emotional Context* | *Unusual Activities* |
| **Collective thought** | | | | | | | | | | | |
| LLaMA-VID | 14.76 | 28.37 | 37.42 | 28.59 | 23.23 | 7.62 | 2.72 | 14.11 | 12.75 | 16.20 | 16.52 |
| GPT-4o | **40.17** | 26.42 | 34.67 | 33.35 | 35.19 | 9.66 | -13.23 | **33.61** | **26.46** | 13.81 | 7.29 |
| Video-ChatGPT | 28.34 | 34.51 | **55.19** | 37.17 | **36.14** | **20.21** | **13.35** | 17.03 | 26.19 | 15.43 | 18.97 |
| mPLUG-Owl-Video | 10.58 | 37.69 | 50.54 | **42.02** | 29.43 | 19.55 | 4.78 | 25.48 | 22.51 | **18.98** | **25.86** |
| Video-LLaVA | 14.18 | **40.36** | 34.82 | 28.71 | 32.62 | 10.23 | 5.99 | 19.27 | 20.28 | 17.14 | 18.98 |
| *Average* | 21.61 | 33.47 | 42.53 | 33.97 | 31.32 | 13.46 | 2.72 | 21.90 | 21.64 | 16.31 | 17.52 |

Table 6: Agreement scores across various visual dimensions between Agent-Debate and collective thought.

| Judge | visual dimensions | | | | | | | | | | |
|---|---|---|---|---|---|---|---|---|---|---|---|
| **& Candidate** | *Cont. & Obj.* | *Fine Action* | *Social Context* | *Visual Context* | *Multi Actions* | *Non-exist (E)* | *Non-exist (NE)* | *Partial Actions* | *Time Order* | *Emotional Context* | *Unusual Activities* |
| **Mixture Judge** | | | | | | | | | | | |
| LLaMA-VID | 29.91 | 48.34 | 54.59 | 29.17 | 37.50 | 7.62 | 2.72 | 14.11 | 23.36 | 12.75 | 16.52 |
| GPT-4o mini | 41.42 | 33.08 | 45.14 | 40.03 | 37.99 | 9.66 | -13.23 | 33.61 | 37.17 | 13.81 | 7.29 |
| Video-ChatGPT | 38.89 | 60.70 | 61.94 | 41.18 | 44.01 | 20.21 | 13.35 | 17.03 | 35.14 | 15.43 | 18.97 |
| mPLUG-Owl-Video | 25.16 | 52.21 | 59.43 | 50.16 | 37.94 | 19.55 | 4.78 | 25.48 | 39.16 | 18.98 | 25.86 |
| Video-LLaVA | 30.29 | 57.66 | 60.05 | 38.08 | 37.92 | 10.23 | 5.99 | 19.27 | 35.64 | 17.14 | 18.98 |
| *Average* | 33.13 | 50.40 | 56.23 | 39.72 | 39.07 | 13.46 | 2.72 | 21.90 | 34.09 | 16.31 | 17.52 |

Table 7: Agreement scores across various visual dimensions between Agent-Debate and mixture judges.

## F   EXTENSION TO THE VIDEOCHATGPT DATASET

We further extended our experiments to include an additional dataset, the *VideoChatGPT* dataset[2]. Due to limited resources, we used half of the data from the *generic* section of this dataset, totaling 1,000 samples.

Table 8: Candidates' Average Scores on the VideoChatGPT Dataset. Each cell shows the average score assigned by the corresponding judge.

| Candidate | Video-LLaVA | InternVL2 | GPT-4o | LLM-Agent-Debate |
|---|---|---|---|---|
| GPT-4o mini | 3.919 | 3.621 | 3.920 | 2.600 |
| Video-ChatGPT | 3.991 | 3.177 | 2.562 | 1.877 |
| mPLUG-Owl-Video | 4.000 | 3.282 | 2.566 | 1.933 |
| Video-LLaVA | 3.997 | 2.898 | 2.863 | 2.086 |

**Findings on Score Assignments.** VLMs such as Video-LLaVA tend to give higher scores to all candidates, mirroring the pattern observed in the CVRR dataset. GPT-4o and InternVL2 provide more

---
[2]https://huggingface.co/datasets/lmms-lab/VideoChatGPT

varied scores, reflecting a more nuanced evaluation. The LLM-Agent-Debate (text-only reference-guided assessment) assigns lower scores across candidates.

Table 9: Score Distributions per Judge (VideoChatGPT Dataset). Each cell shows the total number of occurrences for Ratings 1–5.

| Judge | Rating 1 | Rating 2 | Rating 3 | Rating 4 | Rating 5 |
|---|---|---|---|---|---|
| Video-LLaVA | 5 | 0 | 78 | 3,917 | 0 |
| InternVL2 | 301 | 697 | 778 | 2,171 | 53 |
| GPT-4o | 756 | 776 | 1,078 | 581 | 809 |
| LLM-Agent-Debate | 1,627 | 1,145 | 570 | 421 | 237 |

**Findings on Score Distributions.** Video-LLaVA predominantly assigns a Score 4, indicating a tendency to rate candidates highly regardless of performance. GPT-4o and the LLM-Agent-Debate exhibit a more balanced score distribution, suggesting more critical evaluations. This distribution aligns with the findings in the CVRR data set, reinforcing that less capable VLMs tend to overrate candidates.

Table 10: Weighted Cohen's Kappa Coefficient (%) of Judges with the LLM Agent-Debate method. Higher values indicate stronger agreement.

| Judge | GPT-4o mini | Video-ChatGPT | mPLUG-Owl-Video | Video-LLaVA | Average |
|---|---|---|---|---|---|
| Video-LLaVA | 6.70 | 0.07 | 0.00 | 0.13 | 1.72 |
| InternVL2 | 37.61 | 20.83 | 16.80 | 27.39 | 25.66 |
| GPT-4o | 41.01 | 48.16 | 43.90 | 44.62 | 44.42 |

**Findings on Kappa Agreement.** GPT-4o achieves the highest average Kappa (44.42%), indicating moderate agreement with the LLM Agent-Debate method. Video-LLaVA shows minimal agreement beyond chance, with Kappa scores close to zero on average. These results are consistent with those of the CVRR-ES dataset, indicating that less capable VLMs are unreliable as judges.

## G   ABLATION STUDY ON FINETUNING VIDEO-LLAVA

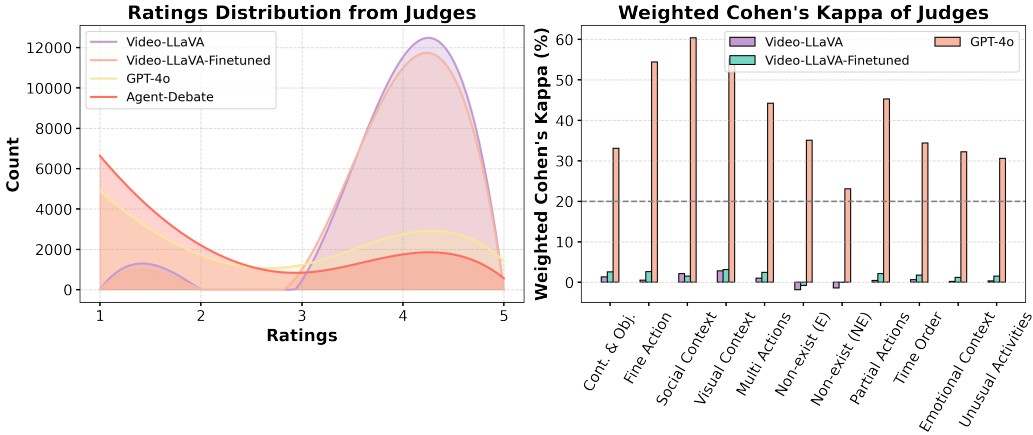

Figure 7: Left: Distributions of Ratings. Right: Judges' *Weighted Cohen's Kappa* values.

In this ablation study, we assess the effect of fine-tuning Video-LLaVA on its evaluation performance. Comparisons of rating distributions and Weighted Cohen's Kappa scores before and after fine-tuning indicate only modest improvement, with ratings still skewed high compared to advanced models such as GPT-4o. These results suggest that fine-tuning alone may not suffice to achieve robust evaluation performance.

