# OpenReview forum: "Is Your Video Language Model a Reliable Judge?"
_ICLR.cc/2025/Conference — ICLR 2025 Poster_

### Official Review · Reviewer_DnJH · 2024-10-27

**Soundness:** 3
**Presentation:** 3
**Contribution:** 2
**Rating:** 8
**Confidence:** 4

**Summary:**

This paper argues that current VLMs, except for GPT-4o, are generally unreliable as judges for video understanding tasks. The study reveals that collective thoughts, which combine judgments from both reliable and unreliable VLMs, do not necessarily improve evaluation accuracy. Even a proposed mixture of judges offers only minor improvements, underscoring the need for more sophisticated methods to account for individual model reliability.

**Strengths:**

- The paper provides detailed experimental results, including both quantitative tables and visual comparisons, which effectively illustrate the trends.
- It is written clearly and easy to follow, making the results seem reproducible.
- The analysis highlights the limitations of relying on collective thought when using unreliable VLMs, contributing to the broader discussion on VLM evaluation.

**Weaknesses:**

- The paper contains several typographical errors, particularly in how different VLM names are written (e.g., inconsistencies with capitalization and dashes).
- The conclusion that VLMs, other than GPT-4o, are unreliable as judges isn't entirely convincing. It would be helpful to provide more insights or potential solutions to address this issue or offer some speculative methods to improve VLMs as judges.
- The use of "hallucination" as a primary explanation for unreliable performance could be expanded with alternative explanations or further justification.
- This paper concludes that VLMs are not yet mature enough to serve as reliable judges for video understanding tasks. However, it lacks a thorough explanation or well-supported evidence to substantiate this claim. The observations and insights presented could be expanded to provide a clearer understanding of why VLMs fall short and what specific factors contribute to their unreliability.

**Questions:**

- Have you considered how the advanced judge processes the original video input and the VLM reviews? Is the final review a consolidation of the VLM assessments or a mixture with the advanced judge’s review as well?
- Could the selection of reliable judges in real-time (inference-time) be achieved without using pre-computed weighted scores (rely on LLM agent debate results, which require reference responses)? It might be useful to explore methods that remove outliers more efficiently and enable VLMs to work independently as judges.
- Could the tendency of VLMs to give high scores be a result of their misunderstanding of the content, rather than an inherent bias towards favorable evaluations?
- Is it possible that the use of Weighted Cohen's Kappa as a metric for selecting reliable judges is not optimal, leading to minimal performance improvements?
- The choice of using the Weighted Cohen's Kappa metric and the specific implementation of collective thought for VLMs as judges are not fully justified. A clearer rationale for the implementation would strengthen the validity of the approach.

---

> ### Author Response · Authors · 2024-11-25
>
> We sincerely appreciate your thorough review and valuable feedback on our paper.  We have revised the submission. Below, we provide detailed responses to each of your points.
>
> ---
>
> **Weaknesses:**
>
> **1. Typographical Errors and Inconsistencies in VLM Names**
>
> Thank you for pointing out these errors. We have revised the manuscript and corrected typographical errors and inconsistencies in the naming of VLMs.
>
> ---
>
> **2. Convincing Conclusion on VLMs' Reliability and Potential Solutions**
>
> We appreciate this insight and agree that our conclusion could be strengthened. In the revised manuscript, we have:
>
> - **Added Additional Experiments:** We conducted further experiments to investigate the relationship between a model's understanding ability and its reliability as a judge. Specifically, we fine-tuned an underperforming model, Video-LLVA, on the CVRR-ES dataset to enhance its understanding ability and evaluated its reliability as a judge. The details can be found in discussions section in paper.
> - **Presented New Findings:** Despite the improved performance of Video-LLAVA after fine-tuning, the model's reliability as a judge showed only a marginal increase. The rating distribution remained skewed toward higher scores, and agreement with the benchmark (measured by the Kappa coefficient) improved minimally.
> - **Provided Insights:** These results suggest that simply enhancing a model's understanding ability is insufficient to make it a reliable judge. Effective evaluation requires both strong comprehension and specific capabilities in critical assessment and reasoning.
> - **Proposed Potential Solutions:** We discuss speculative methods to improve VLMs as judges, such as incorporating specialized training focused on evaluation skills and utilizing robust training to reduce biases and hallucinations.
>
> ---
>
> **3. Expanding on "Hallucination" as an Explanation**
>
> We appreciate your suggestion to delve deeper into the factors contributing to the unreliable judgments by VLMs.
>
> - **Inability to Fully Understand Complex Content:** VLMs may struggle with comprehending  video content that involves nuanced social cues, emotional contexts, or abstract concepts. This limitation leads to superficial understanding and inaccurate evaluations.
> - **Lack of Critical Reasoning Capabilities:** Some VLMs lack the ability to critically analyze and assess the content they process. Unlike humans, they may not question the validity or relevance of certain information.
> - **Inherent Biases in Training Data:** The training data for VLMs may contain biases that skew the evaluations. For example, if the data overrepresent positive feedback, the model might be predisposed to assign higher ratings.
>
> ---
>
> **4. Thorough Explanation of VLMs' Unreliability**
>
> - **Failure to Fully Understand Complex Content:**
>     - **Limited Comprehension:** VLMs judges struggle with videos understanding. Without a deep understanding, their evaluations can be superficial or inaccurate.
>     - **Evidence from Experiments:** We observed instances where VLMs misinterpreted the main events or missed critical details in the videos, leading to incorrect assessments.  These examples can be found in Appendix.
>     - **Uniformly High Ratings:** Our analysis shows that VLMs disproportionately assign higher scores, indicating a tendency to overrate responses regardless of their actual quality. This pattern suggests an inability to discern nuances in performance.
> - **Quantitative Evidence Supporting Unreliability:**
>     - **Low Agreement Scores:** The Weighted Cohen's Kappa scores between VLM judges and the benchmark evaluations are significantly low, indicating poor agreement and reliability. For instance, models like Video-LLAVA and LLAMA-VID have Kappa scores close to zero.
>     - **Rating Distribution Analysis:** We included charts showing that VLMs' ratings are skewed towards higher scores, while more reliable methods like GPT-4o and the agent debate exhibit a more balanced distribution.

---

> ### Author Response · Authors · 2024-11-25
>
> **Questions:**
>
> **1. Processing by the Advanced Judge**
>
> Yes, we have considered the processing mechanism of the advanced judge:
>
> - **Processing Method:** The advanced judge takes both original video input and VLM reviews as inputs. It analyzes the video-question-answer pair independently and also considers the assessments provided by the initial VLM judges. See samples in Appendix.
> - **Final Review Composition:** The final review incorporates the advanced judge's own evaluation of the content and a critical analysis of the VLM assessments.
> - **Rationale:** By leveraging its superior understanding ability, the advanced judge can mitigate the influence of less reliable assessments and enhance the overall evaluation quality.
>
> ---
>
> **2. Real-Time Selection of Reliable Judges Without Pre-Computed Scores**
>
> This is an insightful question. We acknowledge that relying on pre-computed scores and reference responses is not always practical. Some future research
>
> - **Possible Alternative Methods:**
>     - **Consensus-Based Filtering:** Identifying outlier judgments by measuring the deviation of each judge's assessment from the median or mean of all assessments.
>     - **Confidence Scores:** Utilizing the VLMs' own confidence estimates in their evaluations to weigh their contributions.
>     - **Anomaly Detection Algorithms:** Applying statistical models to detect and exclude anomalous judgments without the need for reference responses.
>
> ---
>
> **3. VLMs' High Scores Due to Misunderstanding vs. Bias**
>
> See previous reply in **3. Expanding on "Hallucination" as an Explanation**
>
> ---
>
> **4. Optimality of Weighted Cohen's Kappa for Selecting Reliable Judges**
>
> We agree that the choice of metric is crucial:
>
> - **Justification for Kappa:** We initially selected Weighted Cohen's Kappa due to its ability to measure agreement while accounting for chance agreement and the degree of disagreement.
> - **Limitations Acknowledged:** We recognize that Kappa may not fully capture the nuances of judge reliability, especially in complex, open-ended tasks. We could to use other metrics such as Krippendorff's Alpha.
>
> ---
>
> **5. Rationale for Metric Choice and Collective Thought Implementation**
>
> Here is a clearer rationale:
>
> - **Metric Choice:**
>     - **Rationale:** Weighted Cohen's Kappa was chosen because it accounts for partial agreement and is suitable for ordinal ratings.
>     - **Application:** It allows us to quantify the degree of agreement between judges and the benchmark, aiding in the selection of more reliable judges.
> - **Collective Thought Implementation:**
>     - **Design Explanation:** Our collective thought approach involves aggregating assessments from multiple VLMs to enhance evaluation reliability. We added some samples in Appendix.
>     - **Justification:** The implementation is based on principles of collective thoughts, aiming to leverage diverse perspectives for better outcomes.
>
> ---
>
> **Conclusion**
>
> We are grateful for your constructive feedback, which has significantly helped us improve our paper!

---

> ### Author Response · Authors · 2024-11-25
>
> **Additional Experiments and Insights**
>
> To further strengthen our findings, we conducted additional experiments to explore the relationship between a model's understanding ability and its reliability as a judge. These results including figures can be found in last section in Appendix.
>
> **1. Relationship Between Model Performance and Reliability as a Judge**
>
> We compared several judge models to see if higher performance correlates with greater reliability in evaluation tasks:
>
> | Model | Average Performance(%)[1] | Reliability Score as Judge (%) |
> | --- | --- | --- |
> | GPT-4o | 75.03 | 40.73 |
> | InternVL2 | - | 8.54 |
> | LLAMA-VID | 16.46 | 0.78 |
> | Video-LLAVA | 15.92 | 0.58 |
>
> [1] Results from the CVRR-ES benchmark paper ([Zhang et al., 2024](https://arxiv.org/abs/2405.03690)).
>
> The data shows that GPT-4o, with the highest understanding ability, also has the highest reliability score as a judge. This suggests that a model's understanding ability contributes to its reliability in as judge for evaluation.
>
> **2. Effect of Fine-Tuning on Underperforming Models**
>
> We fine-tuned the underperforming model Video-LLAVA using QLoRA on the CVRR-ES dataset to see if enhancing its understanding ability would improve its reliability as a judge.
>
> **Ratings Distribution:**
>
> | Model | Rating 1 | Rating 2 | Rating 3 | Rating 4 | Rating 5 |
> | --- | --- | --- | --- | --- | --- |
> | Video-LLAVA | 11 | 0 | 456 | 11,530 | 3 |
> | Video-LLAVA-Finetuned | 0 | 0 | 1,005 | 10,984 | 11 |
> | GPT-4o | 4,895 | 1,715 | 1,226 | 2,750 | 1,414 |
> | Agent-Debate | 6,641 | 2,200 | 842 | 1,749 | 568 |
>
> **Average Reliability Scores (Kappa Coefficient):**
>
> | Model | **Reliability Scores(**%) |
> | --- | --- |
> | Video-llava | 0.58 |
> | Video-LLaVA-finetuned | 1.65 |
> | GPT-4o | 40.73 |
>
> Despite fine-tuning, Video-LLAVA's rating distribution remained skewed towards higher ratings, and its reliability as a judge improved only slightly. The agreement scores with the benchmark did not approach those of GPT-4o.
>
> **Conclusion:**
>
> These findings indicate that simply improving a model's understanding ability is not sufficient to enhance its reliability as a judge. A reliable judge must possess both strong comprehension skills and specific capabilities in assessment and critical analysis. In our new submission, we discuss speculative methods to improve VLMs as judges, such as incorporating specialized training focused on evaluation skills and utilizing robust training to reduce biases and hallucinations.
>
>
> ---
>
> We believe these additional experiments provide valuable insights and further address your concerns. Thank you for your constructive feedback, which has helped us improve our work.

---

> ### Comment · Reviewer_DnJH · 2024-11-25
>
> Thanks for addressing all my concerns. The detailed explanations and additional analyses provided valuable insights. Based on this, I raise my score to 8. However, I feel the paper writing still needs improvement, I hope the authors can address this in future versions.

---

> > ### Author Response · Authors · 2024-11-25
> >
> > Thank you for your feedback! We will continue to improve the paper's writing.

---

### Official Review · Reviewer_aECh · 2024-11-03

**Soundness:** 2
**Presentation:** 1
**Contribution:** 3
**Rating:** 6
**Confidence:** 5

**Summary:**

The paper investigates whether Video LLMs (VLMs) can reliably evaluate other VLMs' performance, and explores if using multiple VLMs collectively as judges improves evaluation reliability.

Four different judging methods are studied in the paper:

**Review by LLM Agents Debate (Reference-guided grading)**: providing text LLMs with the reference answer and the generated answer and scoring it
**Review by individual VLM:** A single Video LLM reviews the results from the test model
**Review with Collective Thoughts:** A strong meta-judge Video LLM is provided with scores generated by individual Video LLM judges.
**Mixture of Judges:** A score is generated by weighing the scores predicted by individual Video LLMs on the basis of their performance on the specific question type

The CVRR-ES dataset containing 2,400 question-answer pairs from 217 videos, distributed across 11 different visual dimensions (e.g., multiple actions, social context, emotional context) is used for the evaluation. Weighted Cohen's Kappa is used to measure agreement between different evaluation methods.

The key finding is that current Video LLMs (except GPT4o) are not reliable enough to be used as standalone evaluators. GPT-4o, when used as a sole judge, outperformed its performance when combined with less reliable judges.

**Strengths:**

As many recent works have utilized LLMs and VLMs are judges to evaluate other models, a work exploring the effectiveness of these judges is an interesting topic and of significant potential utility to the community.

**Weaknesses:**

- The paper is poorly structured and written, the different methods studied are not clearly separated and are mixed up in different sections
- The conclusions are based on a single not commonly used dataset, and are unusually strong, its possible that methods like mixture of judges not performing well on this setting is a result of this dataset, and not an universal fact.
- The overall presentation of the data is poor, the paper is full of radar charts with 11 dimensions, however there’s often no real difference in the dimensions
- The quality of the plots is often poor, lacking visual clarity, making it hard to understand the data. I suggest the authors generate the figures in some kind of vector graphics format like pdf instead of generating jpeg images.

**Questions:**

I believe this paper is not quite ready for submission and requires quite a bit more work.

1. Having results on more than one dataset would probably be the single biggest potential improvement.
2. Re-organizing and rewriting methods section of the paper would be a necessity

---

> ### Author Response · Authors · 2024-11-25
>
> Thank you for your valuable feedback. We have revised the submission and addressed your concerns as follows:
>
> ---
>
> **1. Structure and Clarity**
>
> We have reorganized the paper for better clarity:
>
> - **Rewritten Sections:** We have changed the title for each subsections in the section of methodology to make it more clear. We also have a brief description of layout at the beginning of each section.
> - **Improved Writing:** We have revised the writing throughout the paper to enhance readability and coherence. We also updated most of the images for better visualization.
>
> ---
>
> **2. Use of a Single Dataset**
>
> - **Dataset Justification:** We researched multiple VLM benchmarks before made the selection. We chose the CVRR-ES dataset because it is open-ended and covers multiple real-world scenarios like social context, which is not addressed by commonly used, closed-ended datasets, such as Video-MME.
> - **Acknowledged Limitations:** Due to the constraints of limited computation resources, we have moderated our conclusions and acknowledged that findings may not be universally applicable without further validation.
> - **Future Work:** We plan to explore additional datasets in future research to generalize our results.
>
> ---
>
> **3. Data Presentation**
>
> - **Improved Visualizations:** We have updated our figures to include diverse chart types (e.g., bar charts, line plots) for better clarity.
> - **Focused Analysis:** Detailed results are now highlighted.
> - **Enhanced Clarity:** We have ensured that all figures effectively illustrate the key findings.
>
> ---
>
> **4. Quality of Plots**
>
> - **High-Quality Figures:** Most figures have been regenerated in high-resolution formats with improved design for better visualization.
> - **Consistent Formatting:** We have ensured consistency in style and formatting across all figures.
>
> ---
>
> **Additional Comments**
>
> We have made significant revisions to enhance the paper's quality, including restructuring, rewriting, and improving visual presentations. We also conducted ablation study to explore how to improve the reliability of VLM judges.
>
> - **Dataset Limitations:** We found that most existing datasets are closed-ended and not suitable for evaluating open-ended, real-world scenarios.
> - **Focused Scope:** Our chosen dataset aligns with our research objectives, and we have discussed this limitation in the paper.
> - **Future Research:** We aim to include more datasets as they become available to validate and extend our findings.
> - **Methods Section:** The Methodology section has been reorganized with clear subsections for each method, enhancing clarity and readability.
>
> ---
>
> We appreciate your constructive feedback, which has helped us improve our manuscript significantly. We hope that the revised paper addresses your concerns.

---

> > ### Comment · Reviewer_aECh · 2024-11-25
> > **Reviewed the revised version.**
> >
> > Appreciate the attempt at improving the paper, I think the new version is clearly better.  With the changes, I will increase the score, however I think its still marginally below acceptance level. My main reason for concern is that many conclusions can be drawn from this work about effectiveness of Video LLM judges, judge ensembles etc. Doing that on the basis of a single dataset is risky.

---

> ### Author Response · Authors · 2024-11-25
>
> **Additional Experiments and Insights**
>
> To further strengthen our findings, we conducted additional experiments to explore the relationship between a model's understanding ability and its reliability as a judge. These results including the figures that can be found in the last section in Appendix.
>
> **1. Relationship Between Model Performance and Reliability as a Judge**
>
> We compared several judge models to see if higher performance correlates with greater reliability in evaluation tasks:
>
> | Model | Average Performance(%)[1] | Reliability Score as Judge (%) |
> | --- | --- | --- |
> | GPT-4o | 75.03 | 40.73 |
> | InternVL2 | - | 8.54 |
> | LLAMA-VID | 16.46 | 0.78 |
> | Video-LLAVA | 15.92 | 0.58 |
>
> [1] Results from the CVRR-ES benchmark paper ([Zhang et al., 2024](https://arxiv.org/abs/2405.03690)).
>
> The data shows that GPT-4o, with the highest understanding ability, also has the highest reliability score as a judge. This suggests that a model's understanding ability contributes to its reliability in as judge for evaluation.
>
> **2. Effect of Fine-Tuning on Underperforming Models**
>
> We fine-tuned the underperforming model Video-LLAVA using QLoRA on the CVRR-ES dataset to see if enhancing its understanding ability would improve its reliability as a judge.
>
> **Ratings Distribution:**
>
> | Model | Rating 1 | Rating 2 | Rating 3 | Rating 4 | Rating 5 |
> | --- | --- | --- | --- | --- | --- |
> | Video-LLAVA | 11 | 0 | 456 | 11,530 | 3 |
> | Video-LLAVA-Finetuned | 0 | 0 | 1,005 | 10,984 | 11 |
> | GPT-4o | 4,895 | 1,715 | 1,226 | 2,750 | 1,414 |
> | Agent-Debate | 6,641 | 2,200 | 842 | 1,749 | 568 |
>
> **Average Reliability Scores (Kappa Coefficient):**
>
> | Model | **Reliability Scores(**%) |
> | --- | --- |
> | Video-llava | 0.58 |
> | Video-LLaVA-finetuned | 1.65 |
> | GPT-4o | 40.73 |
>
> Despite being fine-tuned, Video-LLAVA's rating distribution remained skewed towards higher ratings, and its reliability as a judge was improved only slightly. The agreement scores with the benchmark did not approach those of GPT-4o.
>
> **Conclusion:**
>
> These findings indicate that simply improving a model's understanding ability is not sufficient to enhance its reliability as a judge. A reliable judge must possess both strong comprehension skills and specific capabilities in assessment and critical analysis. In our new submission, we discuss speculative methods to improve VLMs as judges, such as incorporating specialized training focused on evaluation skills and utilizing robust training to reduce biases and hallucinations.
>
>
> ---
>
> We believe these additional experiments provide valuable insights and further address your concerns. Thank you for your constructive feedback, which has helped us improve our work.

---

> ### Author Response · Authors · 2024-11-28
> **Results on Another Dataset**
>
> Thank you for your positive feedback on the improvements we have made to the paper. We understand your concern about drawing conclusions based on a single dataset. To address this, we have extended our experiments to include an additional dataset, the **VideoChatGPT dataset** (https://huggingface.co/datasets/lmms-lab/VideoChatGPT).
>
> Due to limited time and computational resources, we used half of the data from the generic section of this dataset, totaling **1,000 samples**. Running experiments over **four candidate models** and **four judges** required significant computational effort, including extensive GPU hours and costs associated with running the GPT-4o API.
>
> **Our main results are as follows:**
>
> ### Candidates Received Scores
>
> | **Candidate** | **Video-LLaVA** | **InternVL2** | **GPT-4o** | **LLM** |
> | --- | --- | --- | --- | --- |
> | **GPT-4o mini** | 3.919 | 3.621 | 3.920 | 2.600 |
> | **Video-ChatGPT** | 3.991 | 3.177 | 2.562 | 1.877 |
> | **mPLUG-Owl-Video** | 4.000 | 3.282 | 2.566 | 1.933 |
> | **Video-LLaVA** | 3.997 | 2.898 | 2.863 | 2.086 |
> - **Findings:**
>     - VLMs like **Video-LLaVA** tend to give higher scores to all candidates, mirroring the pattern observed in the CVRR dataset.
>     - **GPT-4o** and **InternVL2**, provide more varied scores, reflecting a more nuanced evaluation.
>     - **LLM** (text-only reference-guided judge) assigns lower scores across candidates.
>
> ### Scores Distribution Given by Judges
>
> | **Judge** | **Rating 1** | **Rating 2** | **Rating 3** | **Rating 4** | **Rating 5** |
> | --- | --- | --- | --- | --- | --- |
> | **Video-LLaVA** | 5 | 0 | 78 | 3,917 | 0 |
> | **InternVL2** | 301 | 697 | 778 | 2,171 | 53 |
> | **GPT-4o** | 756 | 776 | 1,078 | 581 | 809 |
> | **LLM** | 1,627 | 1,145 | 570 | 421 | 237 |
> - **Findings:**
>     - **Video-LLaVA** predominantly assigns **Score 4**, showing a tendency to rate candidates highly regardless of performance**.**
>     - **GPT-4o** and **LLM** exhibit a more balanced score distribution, suggesting a critical evaluation approach.
>     - This distribution aligns with our findings from the CVRR dataset, reinforcing the observation that less capable VLMs tend to overrate candidates.
>
> ### Weighted Cohen's Kappa Coefficient(%) of Judges on Candidates
>
> | **Judge** | **GPT-4o mini** | **Video-ChatGPT** | **mPLUG-Owl-Video** | **Video-LLaVA** | **Average** |
> | --- | --- | --- | --- | --- | --- |
> | **Video-LLaVA** | 6.70 | 0.07 | 0.00 | 0.13 | 1.72 |
> | **InternVL2** | 37.61 | 20.83 | 16.80 | 27.39 | 25.66 |
> | **GPT-4o** | 41.01 | 48.16 | 43.90 | 44.62 | 44.42 |
> - **Findings:**
>     - **GPT-4o** has the highest average Kappa score (**44.42%**), indicating moderate agreement with the LLM reference-guided judge.
>     - **Video-LLaVA** has minimal agreement beyond chance, with average Kappa scores close to zero.
>     - These results are consistent with those from the CVRR dataset, demonstrating that less capable VLMs are unreliable as judges.
>
> **Conclusion:**
>
> - The consistency of results across both the CVRR and VideoChatGPT datasets strengthens our conclusions about the limitations of using less capable VLMs as judges.
> - Our findings suggest that the observed patterns are not specific to a single dataset but are indicative of a broader issue in the evaluation of VLMs.
> - Despite computational constraints, extending our experiments to another dataset enhances the generalizability and robustness of our work.
>
> We hope that these additional experiments address your concern regarding the reliance on a single dataset. Thank you again for your constructive feedback, which has helped us improve the rigor and scope of our paper!

---

> > ### Comment · Reviewer_aECh · 2024-11-29
> >
> > These results are interesting, I would suggest adding the results to final version of the paper. I will increase my final rating to reflect the improvements in this paper.

---

> > > ### Author Response · Authors · 2024-11-30
> > >
> > > Thanks very much for your feedback and valuable suggestions! I will incorporate the results into the final version of the paper as suggested. I truly appreciate your support and constructive comments!

---

### Official Review · Reviewer_zBrV · 2024-11-03

**Soundness:** 3
**Presentation:** 3
**Contribution:** 2
**Rating:** 6
**Confidence:** 4

**Summary:**

This work aims to explore the effectiveness in the evaluation of Video Language Model (VLM), especially focus on collective thinking methods.  Traditional evaluation methods usually rely on a single model, which is susceptible to biases in training data and structure and limiting reliability. This work proposes to take different advanced VLMs and LLMs as judges to evaluate video question-answer pairs. An "evaluation-judgment" framework was implemented, where judges gave scores based on accuracy and relevance on a 1 to 5 scale. The advanced model GPT4o was used to integrate the preliminary review results of VLMs to generate the final evaluation. A mixed judge strategy was used to select the most reliable VLMs to contribute to the final evaluation of each visual dimension based on the weighted Cohen's Kappa score.

**Strengths:**

1. This work is well-written and easy to follow.
2. The experimental results list the performance comparison of different models in various visual dimensions, including social context, emotional context, object instance count, etc. The scores of models such as GPT4o, GPT3.5, Agent Debate, etc. are compared to show their performance in each visual dimension. The performance scores of some models such as Llama-vid, GPT4o-mini, Internvl2, etc. are shown in table form.

**Weaknesses:**

1. There are so many new benchmarks for evaluate the VLM recently, and what is best benchmark for evaluating VLM.
2. This work proposed a new method for evaluating VLM, but this area need an evaluation benchmark(system) rather than just a new method.
3. Overall, even though this work proposes a good method and explore new insight, I think the contribution of this work is not enough for evaluating VLMs.

**Questions:**

Please refer to Weaknesses.

---

> ### Author Response · Authors · 2024-11-25
>
> Thank you for your thoughtful review. Below, we address your concerns point by point. We also have updated most of the charts in our paper for better visualization.
>
> ---
>
> **1. Recent Benchmarks for Evaluating VLMs**
>
> Our research prioritized selecting a dataset suitable for assessing nuanced, open-ended reasoning abilities. While recent benchmarks like **Video-MME** ([Wu et al., 2024] (https://arxiv.org/abs/2405.21075)) offer insights through short-answer, multiple-choice questions, they are less suited for open-ended evaluation. Similarly, **HourVideo** ([Zhang et al., 2024] (https://arxiv.org/abs/2411.04998)) focuses exclusively on long video evaluation, limiting its applicability.
>
> The **CVRR-ES** dataset was chosen for its emphasis on open-ended questions across diverse real-world scenarios, such as social context, emotional understanding, and complex visual reasoning. It challenges VLMs to generate contextually rich responses, making it a robust benchmark for comprehensive evaluation.
>
> ---
>
> **2. Need for an Evaluation Benchmark/System**
>
> Our work emphasizes improving VLM reliability through:
>
> - **A Structured Evaluation Framework:** Assessing VLMs across 11 diverse dimensions to highlight strengths and weaknesses.
> - **Identifying Methodological Gaps:** Addressing limitations in collective intelligence methods that include unreliable models.
> - **Enhancing Judge Reliability:** Exploring key factors such as understanding and critical reasoning to improve VLM evaluation methods.
>
> We view our contributions as a step toward standardized evaluation systems, providing insights that can guide future benchmark development.
>
> ---
>
> **3. Contribution to Evaluating VLMs**
>
> We expanded our analysis to detail VLMs' unreliability, identifying key issues:
>
> - **Limited Understanding Ability:** VLM judges have misinterpreted the events in videos and severe hallucinations (examples in Appendix).
> - **Lack of Critical Review Skills:** VLM judges tend to assign uniformly high ratings, lacking the reasoning to assess response quality critically.
> - **Empirical Evidence of Unreliability:** Low Weighted Cohen’s Kappa scores and skewed rating distributions highlight poor reliability compared to human evaluators.
> - **Contributing Factors:** Limited training diversity and inherited biases hinder VLMs' ability to generalize and assess accurately.
>
> **Key Contributions:**
>
> - Systematic identification of the limitations of VLMs as judges.
> - Empirical evidences supporting the need for improved evaluation methods.
> - Guidance for enhancing VLMs' evaluation capabilities.
> - Proposals for robust evaluation frameworks for complex video understanding tasks.
>
> We hope this detailed response clarifies our contributions and addresses your concerns effectively.

---

> ### Author Response · Authors · 2024-11-25
>
> **Additional Experiments and Insights**
>
> To further strengthen our findings, we conducted additional experiments to explore the relationship between a model's understanding ability and its reliability as a judge. These results including the figures that can be found in the last section in Appendix.
>
> **1. Relationship Between Model Performance and Reliability as a Judge**
>
> We compared several judge models to see if higher performance correlates with greater reliability in evaluation tasks:
>
> | Model | Average Performance(%)[1] | Reliability Score as Judge (%) |
> | --- | --- | --- |
> | GPT-4o | 75.03 | 40.73 |
> | InternVL2 | - | 8.54 |
> | LLAMA-VID | 16.46 | 0.78 |
> | Video-LLAVA | 15.92 | 0.58 |
>
> [1] Results from the CVRR-ES benchmark paper ([Zhang et al., 2024] (https://arxiv.org/abs/2405.03690)).
>
> The data shows that GPT-4o, with the highest understanding ability, also has the highest reliability score as a judge. This suggests that a model's understanding ability contributes to its reliability as judge for evaluation.
>
> **2. Effect of Fine-Tuning on Underperforming Models**
>
> We fine-tuned the underperforming model Video-LLAVA using QLoRA on the CVRR-ES dataset to see if enhancing its understanding ability could improve its reliability as a judge.
>
> **Ratings Distribution:**
>
> | Model | Rating 1 | Rating 2 | Rating 3 | Rating 4 | Rating 5 |
> | --- | --- | --- | --- | --- | --- |
> | Video-LLAVA | 11 | 0 | 456 | 11,530 | 3 |
> | Video-LLAVA-Finetuned | 0 | 0 | 1,005 | 10,984 | 11 |
> | GPT-4o | 4,895 | 1,715 | 1,226 | 2,750 | 1,414 |
> | Agent-Debate | 6,641 | 2,200 | 842 | 1,749 | 568 |
>
> **Average Reliability Scores (Kappa Coefficient):**
>
> | Model | **Reliability Scores(**%) |
> | --- | --- |
> | Video-llava | 0.58 |
> | Video-LLaVA-finetuned | 1.65 |
> | GPT-4o | 40.73 |
>
> Despite being fine-tuned, Video-LLAVA's rating distribution remained skewed towards higher ratings, and its reliability as a judge was improved only slightly. The agreement scores with the benchmark did not approach those of GPT-4o.
>
> **Conclusion:**
>
> These findings indicate that simply improving a model's understanding ability is not sufficient to enhance its reliability as a judge. A reliable judge must possess both strong comprehension skills and specific capabilities in assessment and critical analysis. In our new submission, we discuss speculative methods to improve VLMs as judges, such as incorporating specialized training focused on evaluation skills and utilizing robust training to reduce biases and hallucinations.
>
>
> ---
>
> We believe these additional experiments provide valuable insights and further address your concerns. Thank you for your constructive feedback, which has helped us improve our work.

---

> > ### Comment · Reviewer_yySx · 2024-11-26
> >
> > Thank you for the detailed reply. I don't have any further questions. I'll maintain my original score.

---

> ### Author Response · Authors · 2024-11-28
> **Results on Another Dataset**
>
> Thank you for your time! Please feel free to reach out if you need any further clarification to help increase your score.
>
> Meanwhile, we have extended our experiments to include an additional dataset, the **VideoChatGPT dataset** (https://huggingface.co/datasets/lmms-lab/VideoChatGPT).
>
> Due to limited time and computational resources, we used half of the data from the generic section of this dataset, totaling **1,000 samples**. Running experiments over **four candidate models** and **four judges** required significant computational effort, including extensive GPU hours and costs associated with running the GPT-4o API.
>
> **Our main results are as follows:**
>
> ### Candidates Received Scores
>
> | **Candidate** | **Video-LLaVA** | **InternVL2** | **GPT-4o** | **LLM** |
> | --- | --- | --- | --- | --- |
> | **GPT-4o mini** | 3.919 | 3.621 | 3.920 | 2.600 |
> | **Video-ChatGPT** | 3.991 | 3.177 | 2.562 | 1.877 |
> | **mPLUG-Owl-Video** | 4.000 | 3.282 | 2.566 | 1.933 |
> | **Video-LLaVA** | 3.997 | 2.898 | 2.863 | 2.086 |
> - **Findings:**
>     - VLMs like **Video-LLaVA** tend to give higher scores to all candidates, mirroring the pattern observed in the CVRR dataset.
>     - **GPT-4o** and **InternVL2**, provide more varied scores, reflecting a more nuanced evaluation.
>     - **LLM** (text-only reference-guided judge) assigns lower scores across candidates.
>
> ### Scores Distribution Given by Judges
>
> | **Judge** | **Rating 1** | **Rating 2** | **Rating 3** | **Rating 4** | **Rating 5** |
> | --- | --- | --- | --- | --- | --- |
> | **Video-LLaVA** | 5 | 0 | 78 | 3,917 | 0 |
> | **InternVL2** | 301 | 697 | 778 | 2,171 | 53 |
> | **GPT-4o** | 756 | 776 | 1,078 | 581 | 809 |
> | **LLM** | 1,627 | 1,145 | 570 | 421 | 237 |
> - **Findings:**
>     - **Video-LLaVA** predominantly assigns **Score 4**, showing a tendency to rate candidates highly regardless of performance**.**
>     - **GPT-4o** and **LLM** exhibit a more balanced score distribution, suggesting a critical evaluation approach.
>     - This distribution aligns with our findings from the CVRR dataset, reinforcing the observation that less capable VLMs tend to overrate candidates.
>
> ### Weighted Cohen's Kappa Coefficient(%) of Judges on Candidates
>
> | **Judge** | **GPT-4o mini** | **Video-ChatGPT** | **mPLUG-Owl-Video** | **Video-LLaVA** | **Average** |
> | --- | --- | --- | --- | --- | --- |
> | **Video-LLaVA** | 6.70 | 0.07 | 0.00 | 0.13 | 1.72 |
> | **InternVL2** | 37.61 | 20.83 | 16.80 | 27.39 | 25.66 |
> | **GPT-4o** | 41.01 | 48.16 | 43.90 | 44.62 | 44.42 |
> - **Findings:**
>     - **GPT-4o** has the highest average Kappa score (**44.42%**), indicating moderate agreement with the LLM reference-guided judge.
>     - **Video-LLaVA** has minimal agreement beyond chance, with average Kappa scores close to zero.
>     - These results are consistent with those from the CVRR dataset, demonstrating that less capable VLMs are unreliable as judges.
>
> **Conclusion:**
>
> - The consistency of results across both the CVRR and VideoChatGPT datasets strengthens our conclusions about the limitations of using less capable VLMs as judges.
> - Our findings suggest that the observed patterns are not specific to a single dataset but are indicative of a broader issue in the evaluation of VLMs.
> - Despite computational constraints, extending our experiments to another dataset enhances the generalizability and robustness of our work.
>
> We hope that these additional experiments provide more insights! Thank you again for your constructive feedback, which has helped us improve the rigor and scope of our paper!

---

> > ### Comment · Reviewer_zBrV · 2024-12-02
> > **Thanks for the response**
> >
> > Thanks for the detailed information and new results provided by the authors. The response has addressed most of my concerns.

---

> > > ### Author Response · Authors · 2024-12-02
> > >
> > > Thank you for taking the time to review our work and provide feedback!

---

### Official Review · Reviewer_yySx · 2024-11-03

**Soundness:** 3
**Presentation:** 3
**Contribution:** 3
**Rating:** 6
**Confidence:** 3

**Summary:**

This paper explores an important question in the evaluation of Video Language Models (VLMs): the impact of collective intelligence approaches on evaluation accuracy, especially when combining reliable and less reliable VLMs. This is a relevant topic given the growing reliance on VLMs in various applications where understanding complex video content is essential.

**Strengths:**

Novelty of Approach: The manuscript presents a unique perspective by applying collective intelligence principles to VLM evaluation, attempting to leverage multiple models to mitigate biases present in individual models. This approach is innovative and relevant, as it challenges existing evaluation methodologies that typically rely on single-model assessments.
Analysis of Collective Judgment: The authors' findings regarding the drawbacks of combining reliable and unreliable models are noteworthy. These results caution against blind aggregation in evaluation, highlighting the risk of introducing noise and bias, which can degrade the final evaluation’s reliability.

**Weaknesses:**

While the paper addresses an important topic, the methodology could be more detailed. Readers may benefit from a clearer explanation of how model reliability was determined and how judgments were aggregated. More specifics on the metrics and statistical techniques used would also strengthen the study’s transparency.

**Questions:**

Great work！I have no further questions

---

> ### Author Response · Authors · 2024-11-25
>
> Thank you for your constructive feedback! We appreciate your recognition of the importance of our work. We agree that providing more detail on our methodology will enhance the clarity and transparency of our study.  We also update most charts in our paper for better visualization!
>
> ---
>
> - **Expanded Explanation of Model Reliability Determination:**
>     - **Reliability Metrics:** We have elaborated on how we used the **Weighted Cohen's Kappa** coefficient to assess the reliability of each Video Language Model (VLM) as a judge. This metric measures the agreement between each VLM's evaluations and the baseline established by the LLM agent debate method.
>     - **Calculation Details:** Additional information on how the kappa scores were calculated, including formulas and interpretation guidelines can be found in the our Methodology.
> - **Detailed Description of Judgment Aggregation:**
>     - **Aggregation Process:** We have provided a step-by-step explanation of how judgments from multiple VLMs were aggregated. As shown in figure 2, this includes how the advanced model (GPT-4o) processes the original video input along with the individual VLM reviews to produce a final assessment. We also added some samples in Appendix.
>     - **Collective Thought Method:** See the samples we added in Appendix.
> - **Specifics on Metrics and Statistical Techniques:**
>     - **Statistics:** We have listed all the metrics and statistical techniques employed in our analysis, such as agreement scores, rating distributions. These results can be found in figure 4.
>     - **Transparency in Data Presentation:**  We added more detailed tables, charts, and graphs illustrating the results in the main paper and the Appendix. These visuals are accompanied by explanations to aid understanding.
>
> By providing these additional details, we aim to make our methodology more transparent and accessible to readers. Thank you again for your valuable feedback!

---

> > ### Author Response · Authors · 2024-11-25
> >
> > **Additional Experiments and Insights**
> >
> > To further strengthen our findings, we conducted additional experiments to explore the relationship between a model's understanding ability and its reliability as a judge. These results including figures can be found in last section in Appendix.
> >
> > **1. Relationship Between Model Performance and Reliability as a Judge**
> >
> > We compared several judge models to see if higher performance correlates with greater reliability in evaluation tasks:
> >
> > | Model | Average Performance(%)[1] | Reliability Score as Judge (%) |
> > | --- | --- | --- |
> > | GPT-4o | 75.03 | 40.73 |
> > | InternVL2 | - | 8.54 |
> > | LLAMA-VID | 16.46 | 0.78 |
> > | Video-LLAVA | 15.92 | 0.58 |
> >
> > [1] Results from the CVRR-ES benchmark paper ([Zhang et al., 2024](https://arxiv.org/abs/2405.03690)).
> >
> > The data shows that GPT-4o, with the highest understanding ability, also has the highest reliability score as a judge. This suggests that a model's understanding ability contributes to its reliability in as judge for evaluation.
> >
> > **2. Effect of Fine-Tuning on Underperforming Models**
> >
> > We fine-tuned the underperforming model Video-LLAVA using QLoRA on the CVRR-ES dataset to see if enhancing its understanding ability would improve its reliability as a judge.
> >
> > **Ratings Distribution:**
> >
> > | Model | Rating 1 | Rating 2 | Rating 3 | Rating 4 | Rating 5 |
> > | --- | --- | --- | --- | --- | --- |
> > | Video-LLAVA | 11 | 0 | 456 | 11,530 | 3 |
> > | Video-LLAVA-Finetuned | 0 | 0 | 1,005 | 10,984 | 11 |
> > | GPT-4o | 4,895 | 1,715 | 1,226 | 2,750 | 1,414 |
> > | Agent-Debate | 6,641 | 2,200 | 842 | 1,749 | 568 |
> >
> > **Average Reliability Scores (Kappa Coefficient):**
> >
> > | Model | **Reliability Scores(**%) |
> > | --- | --- |
> > | Video-llava | 0.58 |
> > | Video-LLaVA-finetuned | 1.65 |
> > | GPT-4o | 40.73 |
> >
> > Despite fine-tuning, Video-LLAVA's rating distribution remained skewed towards higher ratings, and its reliability as a judge improved only slightly. The agreement scores with the benchmark did not approach those of GPT-4o.
> >
> > **Conclusion:**
> >
> > These findings indicate that simply improving a model's understanding ability is not sufficient to enhance its reliability as a judge. A reliable judge must possess both strong comprehension skills and specific capabilities in assessment and critical analysis. In our new submission, we discuss speculative methods to improve VLMs as judges, such as incorporating specialized training focused on evaluation skills and utilizing robust training to reduce biases and hallucinations.
> >
> >
> > ---
> >
> > We believe these additional experiments provide valuable insights and further address your concerns. Thank you for your constructive feedback, which has helped us improve our work.

---

> > > ### Comment · Reviewer_yySx · 2024-11-29
> > >
> > > Thank you for the detailed reply. I don't have any further questions. I'll maintain my original score.

---

> > > > ### Author Response · Authors · 2024-11-29
> > > >
> > > > Thank you for your time! Please feel free to reach out if you need any further clarification to help increase your score.

---

### Author Response · Authors · 2024-11-25
**On evaluating strong model using weak model**

Our results indicate that the weaker VLMs, such as Videl-LLaVA, judging stronger models, such as GPT-4o mini, result in unreliable evaluations since these weaker models lack the requisite understanding and critical reasoning abilities. This accords with recent research into weak-to-strong generalization in language models, which proves that if one naively fine-tunes strong models with labels from weaker supervisors, not all of the capabilities of the stronger models are being tapped into [Burns et al., 2023](https://arxiv.org/abs/2312.09390). Equally important is our finding that much stronger models cannot be reliably evaluated by weak VLMs alone. This calls for the development of more sophisticated methods for evaluation to ensure the reliable alignment and performance in such advanced VLMs.

---

### Meta-Review · Area_Chair_Mhrd · 2024-12-20

**Metareview:**

This paper studies how well existing VLMs work as judges of other VLMs in video question answering tasks. Starting from a set of video-question pairs (from CVRR-ES), the authors use a set of existing VLMs to generate a set of answers. Then the authors use a set of VLMs and LLMs as judges to generate reviews of each of the answers (from 1 to 5), then the ratings are combined in a few ways (e.g. mixture of judges, or GPT-4o as a final judge). Evaluation on CVRR-ES shows a range of reliability from candidate judges, and that including reliable and unreliable judges together do not increase reliability. The main conclusion revolves around the use of GPT-4o as a reliable judge. The reviewers appreciated the importance of studying the reliability of VLM judges, and the experimental results on CVRR-ES. However, concerns were raised on the limited choice of datasets and more analysis on model performance and reliability, which authors then added more experiments in their response (e.g. VideoChatGPT dataset). After rebuttal, all reviewers unanimously voted for acceptance. In light of the conclusion, I think it would also benefit the paper for the authors to add a discussion on why not use GPT-4o directly for these tasks (if cost was not a concern), instead of using it as a judge. In the final version of the paper, the authors should note to incorporate all the changes promised to the reviewers in their response.

**Additional Comments On Reviewer Discussion:**

Reviewers initially raised concerns on that all conclusions are drawn from 1 dataset, limited significance of insights from the paper, and presentation issues. Most of the concerns were addressed during the rebuttal (the authors were able to add 1 more dataset, as well as additional analysis on VLM judges). All reviewers voted for accept after the rebuttal. The authors should incorporate all the changes from their rebuttal session into the final version of the paper.

---

### Decision · Program_Chairs · 2025-01-22

Accept (Poster)